# Tumor cell-adipocyte gap junctions activate lipolysis and contribute to breast tumorigenesis

Jeremy Williams [1,2,27], Roman Camarda[1,2,27], Serghei Malkov[3], Lisa J. Zimmerman[4,5], Suzanne Manning[6], Dvir Aran [7,8], Andrew Beardsley[1,9], Daniel Van de Mark[1], Rachel Nakagawa[1,2], Yong Chen [1,10,11,12], Charles Berdan[13,14,15], Sharon M. Louie[13,14,15], Celine Mahieu[1], Daphne Superville[1,2], Juliane Winkler [16,17,18], Elizabeth Willey[16,17], Erica J. Hutchins [1,11], John D. Gagnon[2,19,20], Seda Kilinc Avsaroglu[1], Kosaku Shinoda [1,10,11,21], Matthew Gruner[1], Hiroshi Nishida[22,23], K. Mark Ansel [19,20], Zena Werb [16,17], Daniel K. Nomura [13,14,15], Shingo Kajimura [1,10,11,22,23], Atul J. Butte [7], Melinda E. Sanders [6], Daniel C. Liebler[4,5], Hope S. Rugo [9,24], Gregor Krings[25], John A. Shepherd [26] & Andrei Goga [1,9,17] ✉

A pro-tumorigenic role for adipocytes has been identified in breast cancer, and reliance on fatty acid catabolism found in aggressive tumors. The molecular mechanisms by which tumor cells coopt neighboring adipocytes, however, remain incompletely understood. Here, we describe a direct interaction linking tumorigenesis to adjacent adipocytes. We examine breast tumors and their normal adjacent tissue from several patient cohorts, patient-derived xenografts, and mouse models, and find that lipolysis and lipolytic signaling are activated in neighboring adipose tissue. We find that functional gap junctions form between breast cancer cells and adipocytes. As a result, cAMP is transferred from breast cancer cells to adipocytes and activates lipolysis in a gap junction-dependent manner. We find that connexin 31 (*GJB3*) promotes receptor triple negative breast cancer growth and activation of lipolysis in vivo. Thus, direct tumor cell-adipocyte interaction contributes to tumorigenesis and may serve as a new therapeutic target in breast cancer.

A variety of cancers, including those of the breast, arise within adipose tissue depots[1]. Therefore, heterotypic cell-cell interactions exist between adipocytes and invading cancer cells in these organs during tumor development. We and others discovered that triple-negative breast cancers (TNBC, estrogen/progesterone/HER2 receptor-negative) utilize and require fatty acid oxidation (FAO) to fuel bioenergetic metabolism[2,3]. Epidemiological studies have identified an increased risk for TNBC cancers to form in premenopausal patients with elevated BMI, suggesting that increased adiposity may contribute to tumorigenesis[4]. Nevertheless, how TNBCs acquire fatty acids for tumor growth remains unclear.

Prior work has demonstrated that cancer cells interact indirectly with adipocytes through release of chemokines, cytokines, and other paracrine factors that can remotely activate adipocyte lipolysis. In turn, cancer-associated adipocytes undergoing lipolysis can secrete pro-tumorigenic cytokines, including tumor necrosis factor-α[5–12].

Whether direct interactions at the cancer cell-adipocyte interface contribute to lipolysis in the breast tumor niche, however, is not known.

Multiple studies found that adipocyte-derived fatty acids can be taken up and oxidized by proximate cancer cells[5–11,13]. These studies, however, have widely modeled the cancer-adipocyte interface in vitro using transwell co-culture methods that cannot recapitulate the direct cell-cell contact observed in vivo[7–10,12–14]. Furthermore, clinical evidence for elevated lipolysis in breast tumor-adipocytes has not been well established. Mammary adipocytes undergo enhanced lipolysis when in close proximity to non-tumor epithelial cells, suggesting that local pro-lipolytic mechanisms exist, but have yet to be identified between tumor cells and adipocytes[6,15].

In this study, we investigate the breast tumor-adipocyte interface and examine how cell-cell contact contributes to tumorigenesis. We find diminished lipid content and smaller adipocytes adjacent to patient breast tumors and asked whether this indicates a mechanism of adipocyte lipolysis which depends on tumor cell proximity. Through analysis of patient breast tumors and normal adjacent tissue, direct cancer cell-adipocyte co-cultures, and in vivo xenograft models, we present a mechanism of contact-dependent lipolytic signaling transduced from cancer cells to adipocytes by gap junctions containing connexin (Cx) 31 (*GJB3*). We demonstrate that tumor Cx31 depletion diminishes MYC-high TNBC tumor growth.

## Results

### Diminished lipid content in breast tumor-adjacent adipocytes

To determine if lipolysis occurs in normal tissue adjacent to breast tumors (NAT), which includes adipocytes, we employed four independent strategies. First, we employed three-component breast (3CB) composition measurement, a radiographic imaging method derived from dual-energy mammography that allows for quantification of a tissue's water, lipid, and protein content[16]. We postulated that, if tumors induce lipolysis in adipocytes, we will observe differences in lipid content between normal adjacent tissue (NAT) nearer to the tumor and NAT farther away. Using 3CB imaging, we assessed the lipid content of breast tumors and the first 6 mm of surrounding NAT, segmented into 2 mm "concentric rings" from 46 patients with invasive breast cancer (Fig. 1a and Supplementary Data 1). As we have previously demonstrated[17], we found a significant decrease in lipid content in tumor lesions compared to NAT 0–2 mm away (R1) (Fig. 1b). This difference is congruent with breast tumors being epithelial in nature, while adipose tissue is the major constituent of normal breast[15]. We also found that within NAT there was a significant stepwise decrease in lipid content comparing R3 (4–6 mm) to R2 (2–4 mm), and R2 to R1 (Fig. 1b). In addition, we asked whether changes in lipid content between R3 and R1 NAT correlate with receptor status or tumor grade (Supplementary Data 1 and Supplementary Table 1). We found that NAT surrounding triple-negative (TN) and grade 2/3 tumors trended towards a greater average decrease in lipid content between R3 and R1 than NAT surrounding receptor-positive (RP) and grade 1 tumors, respectively (Supplementary Fig. 1a, b). These data suggest that adipocytes near breast tumors have partially depleted lipid stores, and that TN and higher-grade tumors may induce this phenomenon to a greater degree than RP and low-grade tumors. We quantified average adipocyte size in R1 and R3 in the 11 of the 46 patients imaged with 3CB for whom histological sections of treatment-naïve tumor and NAT at the time of surgical resection were available (Fig. 1a, Supplementary Fig. 1c, and Supplementary Data 1). Similar to the change in lipid content observed with 3CB, we found a significant decrease in adipocyte size in R1 compared to R3 in all patients analyzed, suggesting adipocytes are smaller when nearer to breast tumors (Fig. 1c). Finally, we correlated the change in lipid content and adipocyte size on an individual patient basis. We found a positive trend ($R = 0.5818$, $p = 0.0656$) between the change in lipid content and adipocyte area

(Fig. 1d). Taken together, these data suggest adipocytes are smaller and have diminished lipid content, two phenotypes that are established indicators of lipolysis[18], when adjacent to breast tumors.

### Lipolysis and lipolytic signaling are activated in NAT

Second, we sought to determine if gene expression changes associated with lipolysis were observed in tumor-adjacent adipocytes. We generated a lipolysis gene expression signature by identifying the 100 genes most upregulated when a differentiated adipocyte cell culture model is stimulated with cAMP, an inducer of lipolytic signaling[19]. We then used a publicly available gene expression dataset for primary breast tumors as well as matched NAT 1, 2, 3, and 4 cm away, to determine if enrichment of the lipolysis signature occurred in NAT in comparison to non-tumor breast tissue obtained from healthy individuals using single-set gene set enrichment analysis[20,21]. We found a significant elevation of the cAMP-dependent lipolysis signature in tumor and NAT from all analyzed regions compared to control tissue (Fig. 1e). These data indicate that lipolytic signaling is activated in breast tumor adjacent adipocytes up to 4 cm away from the primary tumor. While adipose tissue is sparsely innervated, a recent study found that adipocytes can propagate pro-lipolytic sympathetic signals via direct transfer of cAMP through adipocyte-adipocyte gap junctions[22]. We observed elevation of cAMP signaling up to 4 cm away from patient tumors (Fig. 1e), suggesting that tumor-adjacent adipocytes might also disperse a pro-lipolytic stimulus to distant adipocytes via gap junctions.

Third, we sought to determine if there are changes to protein abundance in tumor-adjacent NAT indicative of lipolysis activation. We conducted laser capture microdissection (LCM, approximately 10,000 cells per capture) on primary breast tumors from 75 patients, representing all major PAM50 subtypes. For a subset of patients, we also collected matched stroma and/or NAT. As a control, we conducted LCM on non-tumor breast tissue from 42 healthy subjects (Supplementary Data 2a). Global proteomic analysis was performed using liquid chromatography-tandem mass spectrometry (LC-MS/MS) (Supplementary Data 2b). Notably, one of the most significantly upregulated proteins in NAT, and indeed one of the most NAT-specific proteins, compared to all other tissues examined was hepatocyte nuclear factor 4-α (HNF4α) (Fig. 1f). As HNF4α is an established, essential activator of lipolysis in adipose tissue[23], these data indicate lipolysis is robustly activated in breast tumor adjacent adipose tissue.

Fourth, we sought to validate the observations made in our clinical datasets using mouse models of breast cancer. Hormone sensitive lipase (HSL) is a critical lipolytic enzyme; its activation by cAMP-dependent protein kinase A (PKA) leads to phosphorylation at serine 563[18,19], while prolonged activation results in downregulation of total HSL expression through a negative feedback mechanism[24,25]. We performed immunoblot analysis to probe for HSL, phospho-HSL (S563), and HNF4α in tumor and NAT, as well as corresponding control mammary tissues, from three well-characterized breast cancer patient-derived xenograft (PDX) models (HCI002, HCI009, HCI010) and a transgenic model of MYC-driven TNBC (MTB-TOM)[26,27]. In all models analyzed, a downregulation of total HSL in NAT compared to control tissue was observed (Fig. 1g, h). Downregulation of total HSL has been observed in individuals with obesity and in an independent analysis of primary breast tumor NAT, and is thought to be the result of a negative feedback loop in adipocytes in response to chronic lipolysis[24,25]. Additionally, in 3 of the 4 models examined, we found an increase in HNF4α protein or in phospho-HSL/total HSL ratio (Fig. 1g, h), both associated with increased lipolysis[18,23]. Taken together, our concurrent findings in 3 independent clinical datasets and several models of patient-derived xenograft and transgenic mouse breast cancers indicate that lipolysis is activated, to varying degrees, in breast cancer-adjacent adipose tissue. These findings support the conclusion that "normal" tissue adjacent to tumors is, in fact, not normal[28]; in the

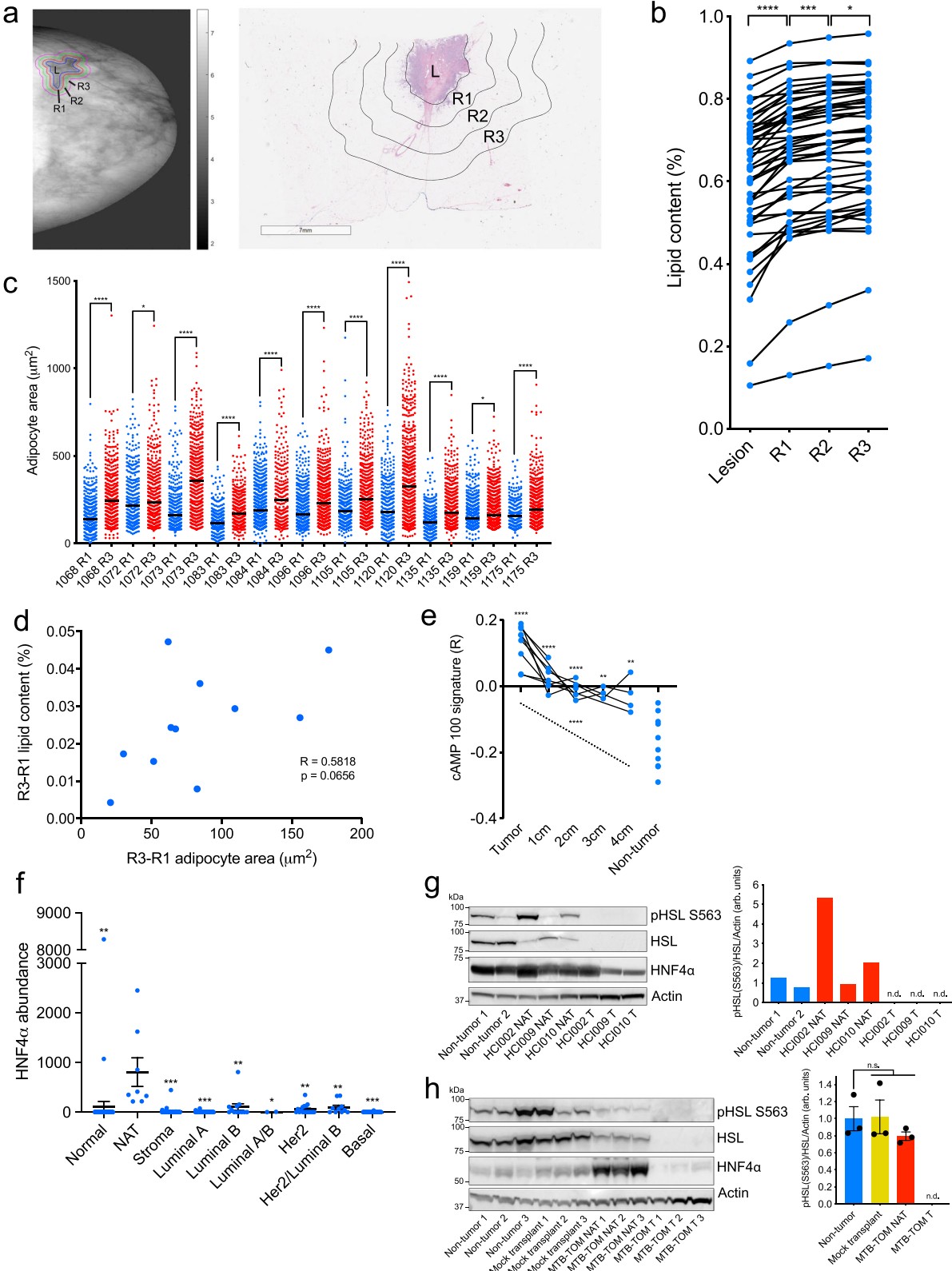

context of breast cancer. Tumor-adjacent adipocytes have markers of activated lipolysis with corresponding diminished lipid stores.

## TNBC-adjacent lipolysis and gap junctions

We next sought to determine the contribution of cell-cell contact to lipolysis activation in breast tumor-adjacent adipocytes. Gap junctions are cell-cell junctions formed by a family of proteins called

connexins, which are known to transport a variety of small molecules (<1 kD), including cAMP[22,29]. Connexins were long thought to play tumor-suppressive roles in cancer[30–32], but recent evidence from a variety of tumor types has challenged this notion[29,33–35]. Given that adipocytes are capable of transferring cAMP and activating lipolysis in a homotypic interaction with other adipocytes[22], we hypothesized that gap junctions may also form between tumor cells and adipocytes

**Fig. 1 | Lipolysis and lipolytic signaling are activated in breast tumor-adjacent adipocytes from breast cancer patients and mouse models of breast cancer.**
**a** Representative lipid content image (left) and hematoxylin and eosin-stained excision specimen (right) from patients with invasive breast cancer. Lesion (L), and NAT 0–2 mm (R1), 2–4 mm (R2), and 4–6 mm (R3) away are indicated. **b** Percent lipid content (lipid content/lipid + water + protein content) of L, R1, R2, and R3 from patients ($n = 46$) with invasive breast cancer (L vs R1 $p < 0.0001$, R1 vs R2 $p = 0.0038$, R2 vs R3 $p = 0.0451$). **c** Adipocyte area in R1 (blue) and R3 (red) from a subset of patients ($n = 11$) in (**b**). Black line indicates mean adipocyte area, and patient identifiers are indicated. Each point represents an individual adipocyte. **d** Correlation of change in lipid content in (**b**) and change in average adipocyte area in (**c**) from R3 to R1 for matched patients in (**c**). **e** ssGSEA enrichment scores for cAMP-dependent lipolysis signature in primary breast tumors ($n = 9$), NAT 1 cm ($n = 7$), 2 cm ($n = 5$), 3 cm ($n = 3$), and 4 cm ($n = 4$), and healthy non-tumor breast tissue ($n = 10$). **f** HNF4α protein abundance from LC-MS/MS of primary healthy control breast tissue ($n = 42$, $p < 0.0001$), NAT ($n = 4$) and stroma ($n = 36$, $p < 0.0001$), and of luminal A ($n = 38$, $p < 0.0001$), luminal B ($n = 6$, $p < 0.0001$), luminal A/B ($n = 1$, $p = 0.0153$), HER2-amplified ($n = 9$, $p < 0.0001$), HER2-amplified/luminal B ($n = 5$, $p < 0.0001$), and basal ($n = 16$, $p < 0.0001$) tumors. Each point represents individual sample LCM on which LC-MS/MS was performed; LCM and LC-MS/MS were performed in ($n = 2$)

technical duplicates on sequential histological slides from each patient, and technical duplicates are displayed. **g** Immunoblot analysis (left) showing expression levels of lipolysis activators HSL and HNF4α, and phosphorylated HSL (pHSL S563) in healthy non-tumor mammary gland and NAT and tumor tissues from a panel of PDXs. Quantification (right) of displayed pHSL/HSL ratio, normalized to b-actin levels, for non-tumor (blue), and NAT (red) and indicated tumors. **h** Immunoblot analysis (left) showing expression levels of lipolysis activators HSL and HNF4α, and phosphorylated HSL (pHSL S563) in healthy non-tumor mammary gland ($n = 3$ mice), mock-transplanted mammary gland ($n = 3$ mice), and NAT and tumor tissues from ($n = 3$) MTB-TOM allografts. Quantification (right) of displayed pHSL/HSL ratio, normalized to b-actin level for each biological replicate. For (**b** and **e**), solid black lines indicate matched samples from individual patients. For (**f** and **h**) mean ± s.e.m. is shown. *$P < 0.05$, **$P < 0.01$, ***$P < 0.001$, ****$P < 0.0001$; repeated measures one-way ANOVA with multiple comparisons (**b**), two-way ANOVA with multiple comparisons (**c**), Spearman correlation and two-tailed $t$ test (**d**), repeated measures mixed effects model with multiple comparisons (**e**), ordinary one-way ANOVA with multiple comparisons (**f** and **h**). For (**g** and **h**), the samples derive from the same experiment, but different gels for pHSL(S563), HNF4α, and Actin, and another for HSL were processed in parallel. Source data are provided as a Source Data file.

---

in a heterotypic fashion to activate lipolysis via transfer of cAMP. Using a well-established dye transfer assay[34], we first probed for presence of functional gap junctions between breast cancer cells. Because gap junction function in breast tumors has not been clearly defined, we tested whether the TNBC cell line HCC1143 or the more indolent RP cell line T47D could transfer gap junction-dependent dyes to the same tumor cell line. Both lines formed functional gap junctions, but dye transfer between HCC1143 cells was 30-fold increased (Fig. 2a) compared to transfer amongst T47D cells. Thus, we reasoned there may be differences in sensitivity to gap junction inhibition between TN and RP cells. Furthermore, given the upregulation of the MYC oncogene in the majority of TNBC[36,37], we asked whether MYC expression affects gap junction dependence. We examined if gap junction inhibition alters cellular ATP as a proxy for cell abundance in a panel of TN and RP human breast cell lines with varying MYC levels[2]. Intriguingly, TNBC cell lines with high MYC expression[2], including HCC1143, were significantly more sensitive to 24 h of treatment with the pan-gap junction inhibitor carbenoxolone (CBX) than the low MYC TNBC or RP cell lines tested (Fig. 2b). In addition, dye transfer to HCC1143 cells was significantly reduced by 30.63% ($p < 0.0001$) following treatment with CBX (Fig. 2c). These data suggest that gap junction communication occurs between breast cancer cells, and that a threshold amount of gap junction activity may be required for MYC-high TN cell viability.

### TNBC tumors feature elevated connexin 31 (*GJB3*)

To delineate the role of connexins in TN compared to RP breast cancer, we examined the expression of the 21 connexin genes in 771 primary human breast cancers, TN ($n = 123$) and RP ($n = 648$), using publicly available RNA-seq data from The Cancer Genome Atlas (TCGA). Of the 20 connexins for which data was available, 5/20 were significantly downregulated, and 11/20 were significantly upregulated. These 11 upregulated connexins included 5 of the 7 gap junction B (GJB) family members (Fig. 2d). To probe gap junction expression at the cellular level, we also examined scRNA-seq ($n = 317$) of primary patient tumors ($n = 11$)[38]. Expression of GJBs was observed in a greater fraction (47.2% vs. 29.8%) of TN than RP tumor cells, and GJBs were the most frequently expressed gap junction family for TN, but not for RP tumor cells (Fig. 2e and Supplementary Fig. 2). As an independent approach to examine in vivo expression of connexins in TNBC, we then performed RNA-seq on MTB-TOM tumors and non-tumor control tissue (Supplementary Data 3). Of the 10 connexins for which data were available, 2/10 were significantly downregulated, 4/10 were significantly upregulated, and 4/10 were not significantly changed in MTB-TOM tumors versus control non-tumor tissue (Fig. 2f). Connexin

31 (*GJB3*, Cx31) was the most significantly elevated connexin in both human TN tumors and the MYC-driven TNBC model. Thus, we focused the remainder of our studies on Cx31. Cx31 has been found to be expressed in keratinocytes, the small intestine, and the colon[39,40]. Although roles for various connexins as oncogenes and/or tumor suppressors have been described[29,33], a pro-tumorigenic function of Cx31 has not been previously established.

### TNBC and mammary adipocytes express Cx31

Accordingly, we sought to determine if functional Cx31-containing gap junctions form between breast cancer cells and adipocytes. To validate the presence of cancer-adipocyte gap junctions in TNBC, we began by examining primary patient biopsies for expression of Cx31 and of pan-cytokeratin to distinguish tumor cells. We found that both TN tumor cells and adipocytes robustly express Cx31 at the plasma membrane. Further, we found many points of cell-cell contact occurred in vivo between tumors and adipocytes (Fig. 3a). To model the cell-cell contact observed in vivo between breast cancer cells and adipocytes, we developed three independent co-culture models. First, we performed 3-dimensional ex vivo studies by co-culturing breast cancer cells directly within primary patient breast fat (Fig. 3b). We stably transduced HCC1143 (TNBC) and T47D (RP) with an mCherry expression plasmid, then introduced either mCherry-HCC1143 or -T47D cells directly into mammary adipose tissue (PT001) and co-cultured overnight. Tumor cell-adipocyte co-cultures were formalin-fixed, paraffin-embedded, and probed for Cx31 and pan-cytokeratin expression, then imaged using immunofluorescent microscopy. We found that both HCC1143 cells and adipocytes robustly expressed Cx31 at the plasma membrane; HCC1143 formed close cell-cell contacts with primary adipocytes (Fig. 3b, top). In contrast, while T47D cells formed cancer cell-cancer cell contacts, we did not observe close cancer cell-adipocyte contacts (Fig. 3b, bottom). These data suggest that Cx31 can be expressed at both the tumor cell and adipocyte plasma membrane, and that breast cancer cells can form close cell-cell contacts with adipocytes.

To determine the role of Cx31 in TNBC-adipocyte interactions, we used CRISPR/Cas9 to generate a series of *GJB3*-depleted TN cell lines (HS578T and HCC1143). In TN MYC-high TN cell line HCC1143, we generated two clones, with -1/3 and -2/3 *GJB3* expression loss (HCC1143 *GJB3*^Med and *GJB3*^Low). In TN MYC-low line HS578T, we generated two distinct clones with -1/3 *GJB3* expression loss (HS578T *GJB3*^Med-1 and *GJB3*^Med-2) (Fig. 3c). Despite several attempts, we were unable to generate TN cell lines with complete Cx31 loss, suggesting that a basal level of Cx31 expression is required for TN cancer cell growth.

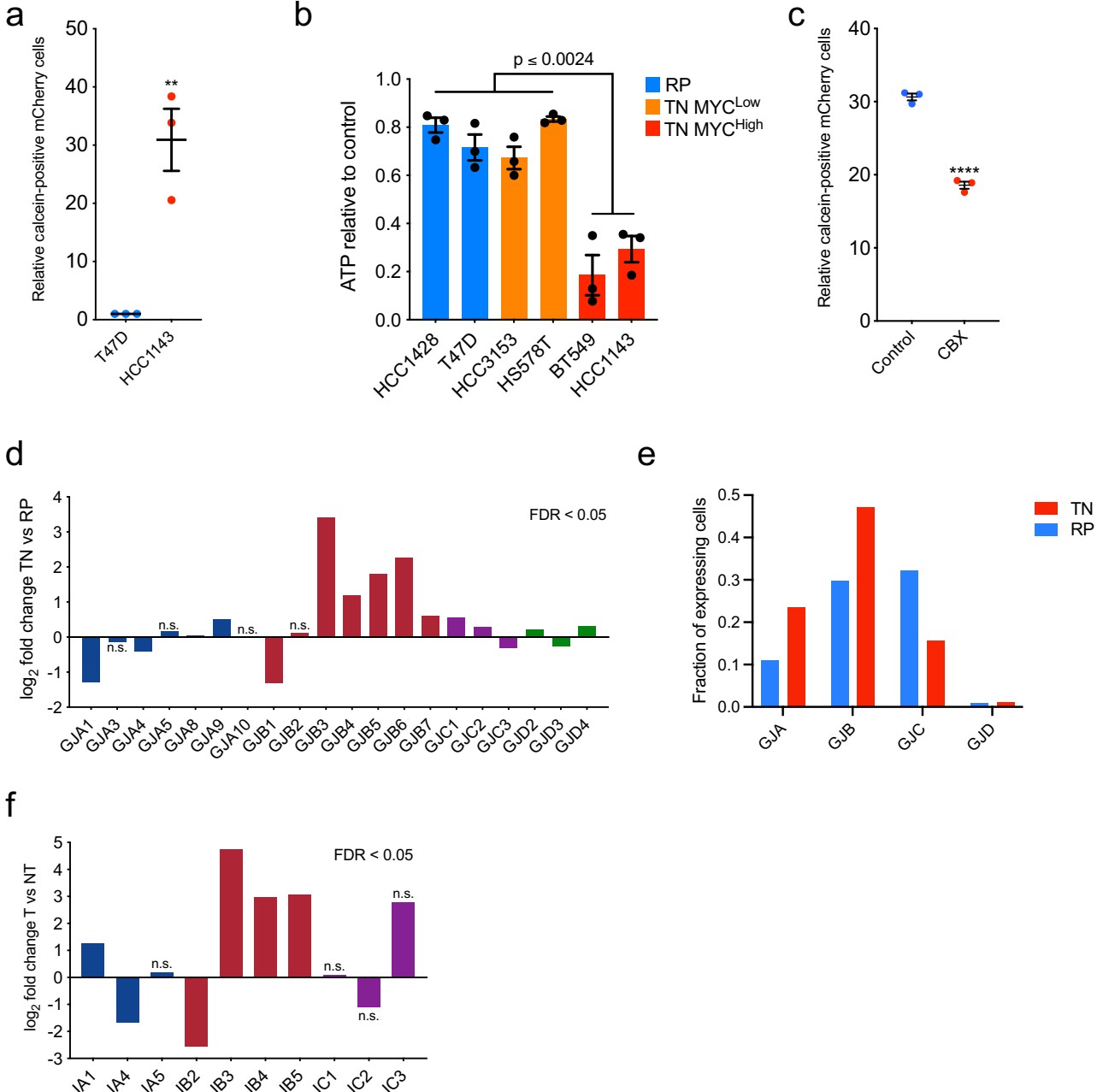

**Fig. 2 | Breast cancer cells form functional gap junctions and express Cx31.**
**a** Relative frequency of dye transfer from Calcein AM-loaded cells (donor) to unloaded mCherry-labelled cells (recipient) as determined by FACS (fluorescence-activated cell sorting) analysis ($p = 0.0050$). Each point represents a biological replicate. **b** ATP levels in TN high MYC (red), TN low MYC (orange), and RP (blue) cell lines after treatment with 150 µM CBX for 24 h, relative to untreated (control) cells. Each point represents a biological replicate averaging three technical replicates. **c** Relative frequency of dye transfer from Calcein AM-loaded cells (donor) to unloaded mCherry-labeled cells (recipient) treated with 150 µM CBX or vehicle control for 24 h, as determined by FACS analysis ($p < 0.0001$). Each point represents a biological replicate. **d** Fold change (log2) in expression of indicated *GJA* (navy), *GJB* (maroon), *GJC* (purple), and *GJD* (green) connexin genes in TN ($n = 123$)

versus RP ($n = 648$) tumors based on RNA-seq data acquired from TCGA of 771 breast cancer patients. **e** Fraction of cells in patient tumors of RP (blue, $n = 6$) and TNBC (red, $n = 5$) subtypes expressing indicated gap junction (GJ) family members, based on sc-RNA-seq of 317 tumor cells. **f** Fold change (log2) in expression of indicated *GJA* (navy), *GJB* (maroon), and *GJC* (purple) connexin genes in tumor (T, $n = 10$) versus non-tumor (NT, $n = 3$) tissues based on RNA-seq data from MTB-TOM allograft-bearing mice or healthy controls, respectively. For (**a**–**c**) mean ± s.e.m. of three independent biological replicates is shown. **P < 0.01, ****P < 0.0001; unpaired two-tailed $t$ test (**a** and **c**); ordinary one-way ANOVA with multiple comparisons (**b**). For (**d** and **f**), all differential expression analysis was done using the "limma" R package with a 0.05 adjusted $P$ value. Source data are provided as a Source Data file.

## Cx31 depletion impacts TNBC-adipocyte cell contact

To examine how Cx31 expression impacted cancer cell-adipocyte contact, we performed ex vivo co-cultures with primary patient breast fat using the partially depleted Cx31 cell lines. We stably transduced TN HCC1143 *GJB3*[WT] and *GJB3*[Low] cell lines, as well as RP line T47D, with a GFP expression plasmid, then cultured each line directly within primary mammary adipose tissues from healthy individuals (PT002, PT003). After overnight incubation, co-cultured tissues were formalin-fixed and probed for expression of Cx31 and lipolysis marker pHSL(S563)[18]. Tissues were then cleared[41] and imaged via whole mount

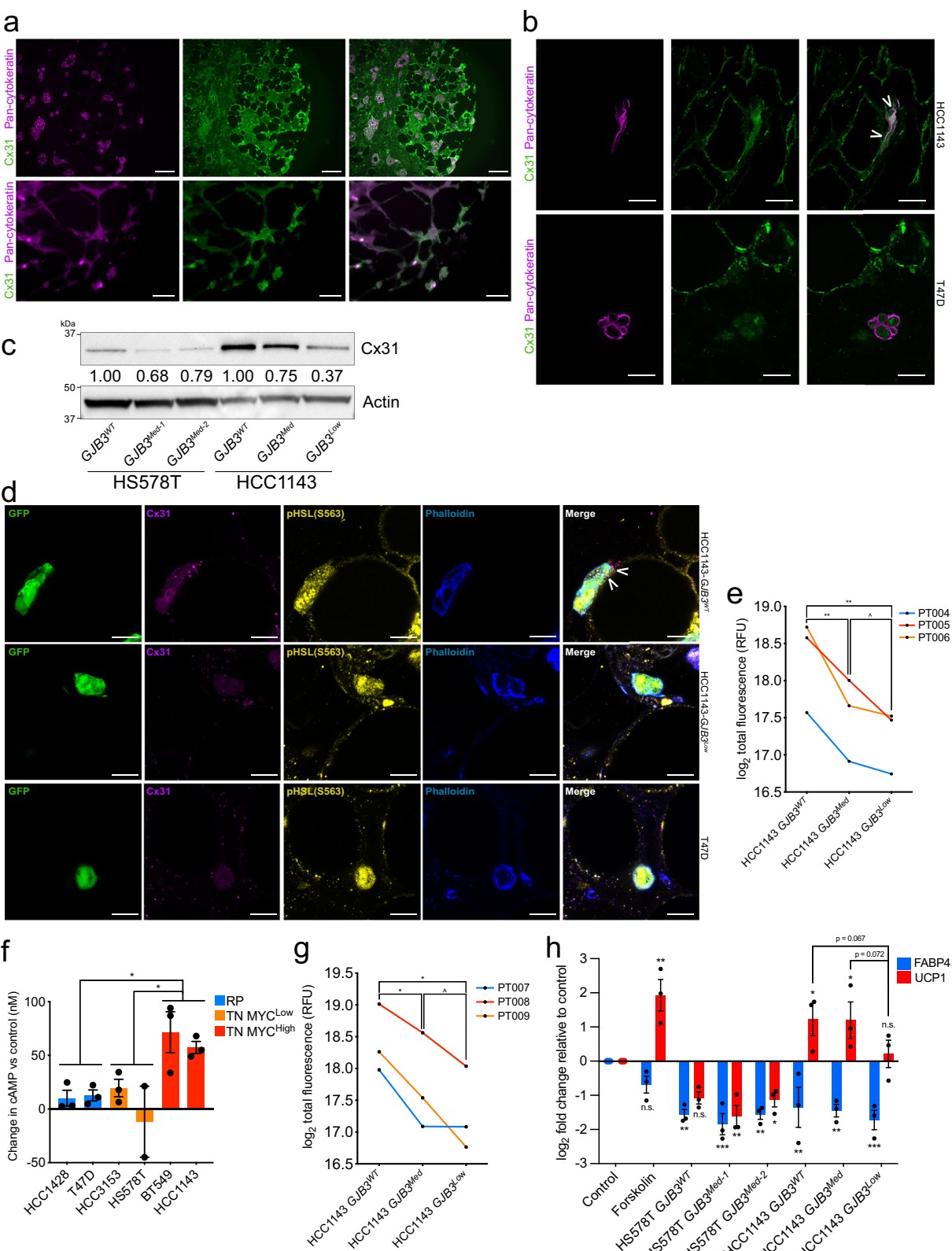

fluorescence microscopy. We found that HCC1143 *GJB3*[WT] cells formed extended cancer cell-adipocyte contacts, in tight conformation with adjacent adipocytes (Fig. 3d and Supplementary Fig. 3a, top). GJB3 aggregates forming puncta at the cancer cell-adipocyte interface were observed. In contrast, Cx31-depleted HCC1143 *GJB3*[Low] cells formed tangential contacts with adjacent adipocytes (Fig. 3d and Supplementary Fig. 3a, middle), which we note closely mimic the tangential

cancer cell-adipocyte conformation observed in T47D (RP) co-cultures (Fig. 3d and Supplementary Fig. 3a, bottom). Both HCC1143 *GJB3*[Low] (TN) and T47D (RP) co-cultures lacked GJB3 puncta at the cancer cell-adipocyte interface. In mock co-cultures, our positive control for-skolin, which raises intracellular cAMP levels by activating adenylyl cyclase[19], robustly induced pHSL(S563) expression and increased puncta compared to vehicle-treated mammary adipose tissue

**Fig. 3 | Breast cancer cell-adipocyte gap junctions form, transfer cAMP, and activate lipolytic signaling dependent on Cx31 expression. a** Staining with Cx31 (green) and pan-cytokeratin (magenta) of primary TNBC patient biopsies. Scale bar, top 100 μm, bottom 25 μm. **b** Staining with Cx31 (green) and pan-cytokeratin (magenta) of primary mammary adipose tissue from a healthy individual (PT001) injected with TN mCherry-HCC1143 cells (top) or RP mCherry-T47D cells (bottom) and co-cultured overnight. White arrowheads indicate staining of Cx31 along point of contact between HCC1143 and adipocyte plasma membranes. Scale bar, 25 μm. **c** Immunoblot analysis showing protein expression levels of Cx31 in vitro in (**a**) panel of clonally derived control *GJB3*^WT and partial depletion TN lines with one-third and two- thirds loss of *GJB3* expression. For the Cx31-depleted lines each clone is referred to by level of *GJB3* expression (e.g., *GJB3*^Med expresses two-thirds WT level, and *GJB3*^Low expresses one third *GJB3*^WT level). Quantification of displayed Cx31 level normalized to b-actin level is indicated. **d** Staining with Cx31 (magenta), pHSL(S563) (yellow), and phalloidin (blue), of healthy patient primary mammary tissue (PT002) injected with GFP-expressing HCC1143 *GJB3*^WT (top), HCC1143-*GJB3*^Low (middle), or T47D cells (bottom) and co-cultured overnight. White arrowheads indicate Cx31 staining at GFP cancer cell-adipocyte interface. Scale bar, 20 μm. **e** Dye transfer from indicated HCC1143 control and Cx31-depleted lines to primary mammary adipose tissue of indicated ($n = 3$) healthy individuals. **f** cAMP levels in TN high MYC (red), TN low MYC (orange, $p = 0.0487$), and RP (blue, $p = 0.487$) cell lines after treatment with 150 μM CBX for 24 h, relative to untreated (control) cells. Each point represents a biological replicate averaging three technical replicates. **g** cAMP transfer from indicated HCC1143 control and Cx31 partial expression loss lines to primary mammary adipose tissue of indicated ($n = 3$) healthy individuals. **h** Fold change in *UCP1* (red) and *FABP4* (blue) expression in differentiated adipocytes after treatment with vehicle (control) or 10 μM forskolin, or co-cultured with indicated Cx31 partial expression loss lines for 24 h. Representative results from experiments done in biological triplicates shown for (**a**–**d**). For (**f** and **h**) mean ± s.e.m. of three independent biological replicates is shown. ^$P < 0.10$, *$P < 0.05$, **$P < 0.01$, ***$P < 0.001$; repeated measures one-way ANOVA with multiple comparisons for (**e** and **g**), ordinary one-way ANOVA with multiple comparisons for (**f** and **h**). Source data are provided as a Source Data file.

(Supplementary Fig. 3b–d). We observed greater pHSL(S563) expression and elevated puncta in adipose tissue co-cultured with HCC1143 *GJB3*^WT cells than tissues with HCC1143 *GJB3*^Low or T47D cells (Supplementary Fig. 3e, f), indicating more cAMP-dependent PKA activity. These results suggest that Cx31 level in breast cancer can mediate cell contact with surrounding adipocytes and alter lipolytic signaling.

## TNBC cells form functional GJB3 gap junctions with adipocytes

We next sought to determine if Cx31 expression impacted tumor cell-adipocyte communication using a co-culture model in which HCC1143 *GJB3*^WT, *GJB3*^Med, or *GJB3*^Low cells were seeded in 2D culture and loaded with gap junction-transferable dye. We added primary mammary adipose tissue from three healthy individuals (PT004, PT005, PT006) directly on top of the monolayers to permit direct contact. Tumor cells and adipocytes were co-cultured for 5 h and then assayed for dye transfer from the cancer cells to adipocytes. We found that robust dye transfer occurred from the HCC1143 *GJB3*^WT cells to mammary adipocytes from all three patients (Fig. 3e). However, depletion of Cx31 expression by 1/3 or 2/3 in the *GJB3*^Med and *GJB3*^Low lines, respectively, resulted in a significant decrease in dye transfer compared to *GJB3*^WT control cells (Fig. 3e). These data suggest that functional gap junctions form between TN breast cancer cells and adipocytes and can be diminished by Cx31 depletion.

## Breast cancer cell gap junctions are permeable to cAMP

To determine if breast cancer cell gap junctions are permeable to cAMP, we treated a panel of human TN and RP cell lines with CBX for 24 h to inhibit pan-gap junction function and ascertain if cAMP was retained in the tumor cells. In 5 of 6 lines tested, we found marked increases in the levels of intracellular cAMP concentration in CBX- versus vehicle-treated cells (Fig. 3f). Additionally, significantly higher concentrations of cAMP were observed in high MYC TN cells in comparison to low MYC TN or RP cells (Fig. 3f). The increase in intracellular cAMP following pan-gap junction inhibition in 5 of 6 lines examined suggests that breast cancer cell gap junctions are indeed permeable to cAMP.

## GJB3 gap junctions transduce cAMP from TNBC to adipocytes

We next tested whether cAMP is directly transferred from breast cancer cells to adipocytes and if the abundance of Cx31 alters transfer. HCC1143 *GJB3*^WT, *GJB3*^Med, or *GJB3*^Low cells were seeded and loaded with a fluorescent cAMP analogue (fluo-cAMP). These monolayer cultures were then co-cultured in direct contact with primary mammary adipose tissue from three healthy individuals (PT007, PT008, PT009) and incubated for 5 h. Adipocytes were then isolated from the tumor cells and assayed for fluo-cAMP. We found that cAMP transfer occurred from control *GJB3*^WT cells to adipocytes from all three patients (Fig. 3g).

However, as we observed with transfer of gap junction-permeable dye (Fig. 3e), depletion of Cx31 resulted in a significant reduction of cAMP transfer (Fig. 3g). Thus, cAMP is transferred from TN breast cancer cells to adipocytes and is diminished following Cx31 depletion.

## TNBC-adjacent adipocytes exhibit activated cAMP signaling

We next sought to determine if downstream cAMP signaling is activated in adipocytes in a gap junction-dependent manner. To determine if cAMP signaling is activated in adipocytes upon cell-cell contact with breast cancer cells, we used a primary mouse preadipocyte model that can be differentiated to adipocytes in vitro[19,42]. This model is ideal to study downstream signaling during co-culture because changes in adipocyte transcription can be assayed via qRT-PCR using murine-specific primers. Adipocytes were terminally differentiated and then HS578T and HCC1143 *GJB3* partial depletion cell lines were seeded directly on top of adipocyte cultures. After co-culturing the cells for 24 h, we extracted RNA and assayed for changes in murine-specific (thus adipocyte-specific in this system) expression of *UCP1*, a known cAMP-responsive gene in adipocytes[19], to measure cAMP signaling. We also assayed for mouse *FABP4* expression as a marker of adipocyte differentiation. Our positive control, forskolin, robustly induced *UCP1* expression compared to vehicle-treated cells (Fig. 3h). Co-culturing with HCC1143 *GJB3*^WT and *GJB3*^Med lines both induced adipocyte UCP1 expression, but *UCP1* induction was significantly reduced in the *GJB3*^Low co-cultures (Fig. 3h). In contrast, none of the MYC low HS578T lines, including the *GJB3*^WT control, were capable of inducing adipocyte *UCP1* expression (Fig. 3h). All conditions, including forskolin treatment, resulted in reduced *FABP4* expression (Fig. 3h), suggesting effects on adipocyte differentiation are distinct from those observed on cAMP signaling. Given that Cx31 expression is similar in HS578T *GJB3*^WT and HCC1143 *GJB3*^Low cells (Fig. 3c), and that neither activate cAMP signaling (Fig. 3h), it is possible that a Cx31 expression threshold is required for breast cancer cells to activate cAMP signaling in adjacent adipocytes. Although direct transfer of cAMP amongst adipocytes via a homotypic gap junction interaction has been described[22], gap junction-dependent activation of adipocyte lipolysis in a heterotypic manner by a tumor cell has not been previously demonstrated.

## Cx31 gap junctions promote breast tumorigenesis in vivo

Finally, we sought to determine the contribution Cx31 gap junction expression to tumorigenesis. We found that HS578T *GJB3*^Med-1 and *GJB3*^Med-2, and HCC1143 *GJB3*^Med cell lines did not display a difference in proliferation compared to their respective *GJB3*^WT control lines (Fig. 4a). In contrast, HCC1143 *GJB3*^Low cells demonstrate a significant reduction in proliferation, while maintaining 93.7% viability relative to Cas9 controls (Fig. 4a). These data suggest that, even in the absence of breast cancer cell-adipocyte interaction, Cx31 promotes breast cancer

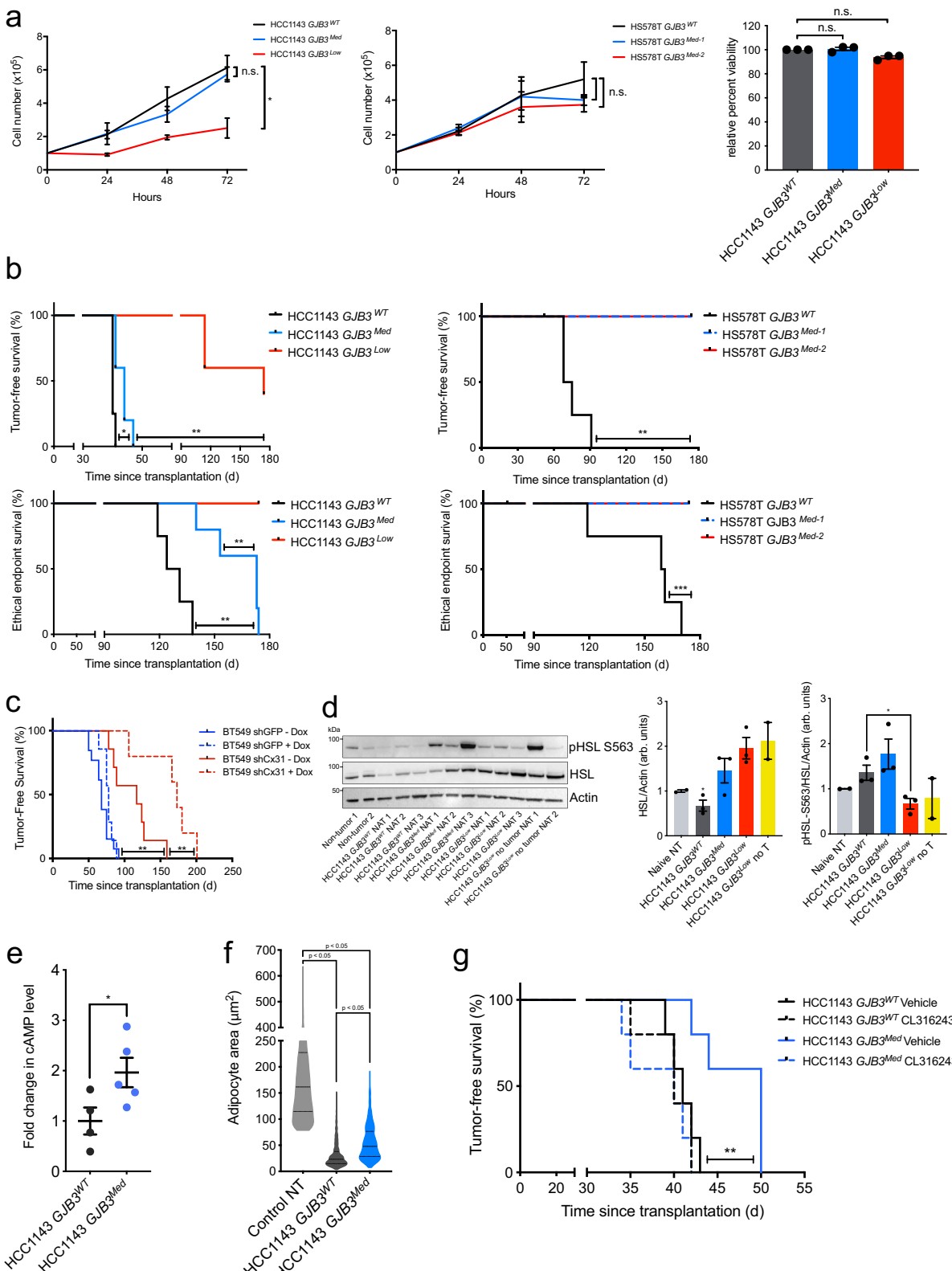

cell proliferation. To determine the contribution of Cx31 to breast tumorigenesis in vivo, we transplanted each of the HS578T and HCC1143 Cx31 partial depletion lines into the mammary fat pad of immune-compromised NOD-SCID/gamma (NSG) female mice and assayed for time of tumor onset and ethical endpoint (when the tumor reaches 2 cm in any dimension). Remarkably, the HS578T lines, in which partial *GJB3* knockout had no effect on cell proliferation ex vivo

(Fig. 4a), 0/10 mice that received HS578T *GJB3*^Med-1 or *GJB3*^Med-2 xenografts (5 per line) developed tumors within 180 days (Fig. 4b). Among the HCC1143 lines, the *GJB3*^Med line displayed a significant delay in both tumor onset and time to ethical endpoint, while only 3 of 5 mice transplanted with the *GJB3*^Low line developed tumors and none reached ethical endpoint within 180 days (Fig. 4b). We performed an independent xenograft model wherein inducible Cx31 hairpins were

**Fig. 4 | Cx31 loss impairs breast cancer cell growth in vitro, tumorigenesis, and activation of lipolysis in adjacent adipocytes in vivo. a** Cell growth of indicated Cx31 partial depletion cell lines in HCC1143 (left) and HS578T (middle) over 72 h ($n = 3$ biological replicates), and cell viability at 72 h of indicated lines normalized to WT control (right, $n = 3$ biological replicates). **b** Kaplan–Meier analysis of tumor onset (top) and ethical endpoint survival (bottom) of mice bearing indicated Cx31 partial expression loss orthotopic xenografts ($n = 5$ per group). **c** Kaplan–Meier analysis of tumor onset in mice bearing indicated orthotopic xenografts with inducible Cx31 (shCx31) or GFP (shGFP) hairpin, with doxycycline (solid line, shGFP $n = 7$, shCx31 $n = 5$ mice) and without doxycycline (broken line, shGFP $n = 13$, shCx31 $n = 5$ mice). **d** Immunoblot analysis (left) showing expression levels of HSL and phosphorylated HSL (pHSL S563) in healthy non-tumor mammary gland (gray, $n = 2$ mice) and NAT from mice bearing indicated *GJB3* WT (black), Med (blue) or Low (red) xenografts ($n = 3$) or mice that were transplanted, but did not develop a tumor (yellow, $n = 2$). Quantification of displayed total HSL(middle) and pHSL/HSL ratio (right), normalized to b-actin levels. Biological replicates from distinct mice are indicated. The samples derive from the same experiment, but different gels for HSL and Actin, and another for pHSL, were processed in parallel. **e** Fold change in cAMP levels in HCC1143 *GJB3*^Med ($n = 5$) xenografts versus HCC1143 *GJB3*^WT ($n = 4$) xenografts ($p = 0.0492$). **f** Adipocyte area adjacent to HCC1143 *GJB3* Med xenografts (pooled from $n = 5$ tumors, $n = 517$ adipocytes) and HCC1143 *GJB3* WT xenografts (pooled *from* $n = 4$, $n = 771$ adipocytes) and area in control non-tumor (NT) tissue (pooled $n = 3$ mice, $n = 2611$ adipocytes). Broken line indicates mean adipocyte area; dotted lines indicate quartiles. Each point represents an individual adipocyte. **g** Kaplan–Meier analysis of tumor onset of mice bearing HCC1143 *GJB3*^WT (black) or *GJB3*^Med (blue) orthotopic xenografts ($n = 5$ per group) and treated with vehicle (solid line) or with 1 mg/kg CL316243 (broken line). For (**b** and **c**), ethical endpoint survival indicates the percentage of mice bearing xenografts <2 cm in any dimension. For (**a**, **d**, and **e**) mean ± s.e.m. is shown. *$P < 0.05$, **$P < 0.01$, ***$P < 0.001$, ****$P < 0.0001$; unpaired two-tailed $t$ test (**a**) (left and center) and (**e**), log-rank test (**b**, **c**, and **g**), ordinary one-way ANOVA with multiple comparisons (**a**) (right), (**d** and **f**). Source data are provided as a Source Data file.

transduced into the TN-MYC^High BT549 human breast cell line and found that Cx31 depletion significantly enhanced tumor-free survival compared to controls (Fig. 4c). Our data indicate that decreasing Cx31 expression is sufficient to impair tumor growth, suggesting that Cx31 gap junctions promote breast tumorigenesis in vivo.

### NAT of GJB3-depleted tumors exhibit decreased lipolysis
We sought to clarify the effects of Cx31 on lipolysis versus other effects on tumor growth. To determine if control and Cx31 partially depleted tumors differentially induced lipolysis, we collected tumor and NAT from HCC1143 *GJB3*^WT, *GJB3*^Med, and *GJB3*^Low tumor-bearing mice, as well as residual mammary glands from the two *GJB3*^Low mice that were transplanted but never developed tumors. Using immunoblot analysis, we probed for markers of lipolysis. Notably, a marked reduction in total HSL expression was found in 3 of 3 HCC1143 *GJB3*^WT NAT samples compared to control tissues (Fig. 4d), consistent with persistent activation of lipolysis leading to HSL downregulation[24,25]. In contrast, we did not observe a consistent change in HSL expression in any of the other NAT samples analyzed from tumors with partial Cx31 expression loss (Fig. 4d). Interestingly, we found a marked increase in phospho-HSL/HSL ratio in both the HCC1143 *GJB3*^WT and *GJB3*^Med NAT samples, but this difference was significantly reduced in HCC1143 *GJB3*^Low NAT (Fig. 4D). The increase in phospho-HSL/HSL in *GJB3*^Med NAT may be due to alternative modes of lipolysis activation, such as secreted pro-lipolytic cytokines[5], which is congruent with the observed increase in *UCP1* expression during *GJB3*^Med-adipocyte co-culture (Fig. 3h). To further interrogate lipolytic signaling in NAT, we probed for cAMP abundance in HCC1143 *GJB3*^WT and *GJB3*^Med tumors by mass spectrometry. We found a significant increase in intratumoral cAMP level in HCC1143 *GJB3*^Med tumors compared to the *GJB3*^WT control tumors (Fig. 4e), consistent with diminished transfer of cAMP to NAT. We examined *GJB3*^WT and *GJB3*^Med tumors and associated NAT, and assayed for differences in adjacent adipocyte size, as an indicator of lipolysis. We found a significant increase in the average size of adipocytes adjacent to *GJB3*^Med tumors compared to *GJB3*^WT control tumors (Fig. 4f), again supporting a decreased induction of lipolysis in NAT from Cx31 partial knockout tumors.

### Activation of lipolysis rescues GJB3-depleted tumor growth
Finally, if the delay in HCC1143 *GJB3*^Med tumor onset (Fig. 4b) was due to an inability to activate lipolysis in adjacent adipocytes, we reasoned that pharmacological activation of lipolysis should rescue this phenotype. Indeed, we found that daily intra-peritoneal injection of CL316243, a specific β3-receptor agonist known to activate lipolysis in vivo[43], rescued the delay in tumor onset observed in HCC1143 *GJB3*^Med tumors, but did not further promote the growth of HCC1143 *GJB3*^WT tumors (Fig. 4g). Taken together, these data indicate that cAMP

signaling and lipolysis are activated in breast tumor-adjacent adipocytes, and that the abundance of Cx31 expression alters these phenotypes in vivo.

## Discussion
Here we find that lipolysis is activated in breast cancer-adjacent adipose tissue, and that functional gap junctions form between breast cancer cells and adipocytes. cAMP can be transferred via breast cancer and activate lipolysis in adjacent adipocytes. Higher Cx31 expression is associated with increased cAMP transfer and induction of lipolysis, as well as more aggressive tumor growth. We established a previously unappreciated, functional role for GJB3 gap junctions in activating lipolysis in tumor-adjacent adipose tissue and promoting breast tumor growth in vivo. Breast cancer-associated gap junctions represent a largely unexplored therapeutic target to treat breast tumors. The recent discovery of gap junction formation and pro-tumorigenic signal exchange between brain metastatic carcinoma cells and astrocytes[34] suggests that gap junction-dependent interactions between tumor and non-tumor cells may be an emerging hallmark of tumorigenesis.

In the breast microenvironment, a diversity of pro-tumorigenic interactions occur between cancer cells and adipocytes, including the transfer of lipolytic factors to adipocytes by tumor cells. Most previously established mechanisms of lipolysis induction rely on remote signaling to adipocytes[44,45], either through circulatory endocrine factors that activate β-adrenergic receptors on the adipocyte membrane, or through paracrine cytokines, chemokines, and growth factors secreted from nearby adipocytes. Models recapitulating tumor-adipocyte interactions often do not consider contact-mediated signaling, and thus the contributions of direct, juxtacrine tumor-adipocyte signaling to tumorigenesis.

In this study, we observe direct transfer of lipolytic cAMP signaling and contributions to TNBC tumorigenesis by GJB3 gap junctions, a potential link between dependance on fatty acid oxidation observed in TNBC[2] and the adipose-rich breast tumor niche. The present data do not, however, exclude transfer of other factors by these GJB3 gap junctions, or transfer of cAMP to adipocytes by other kinds of gap junctions. The present data also do not indicate that intratumor cAMP levels are greater in TN compared to RP tumors, but that elevated Cx31 in TNBC permits increased transfer of cAMP signal from tumor to adipose tumor microenvironment and increased lipolysis in tumor-adjacent adipocytes. Fully isolating the contribution of Cx31 to this phenotype presents a technical challenge. The inability to generate fully Cx31-null patient-derived TNBC cell lines suggests that some degree of expression is required to maintain tumor cell proliferation. The prolonged tumor-free survival and time to ethical endpoint observed for mice bearing *GJB3*-depleted TN-MYC^Low xenografts, which in vitro were less sensitive to gap junction inhibition than

TN-MYC[High] cell lines, imply an additional tumor-intrinsic role for Cx31 as well. Several *GJB* family members are differentially increased when comparing transcript levels in patient TN to RP tumors, suggesting multiple gap junction proteins may be relevant to tumor growth in vivo.

Further exploration of juxtacrine signaling, in particular GJ-mediated intercellular communication, may be critical in understanding the tropism of metastatic cancers to adipose-rich tissue depots throughout the body. Aggressive ovarian and gastric cancers share tropism for the omentum, a major adipose tissue depot, and also feature pro-metastatic roles for tumor-adjacent fat[7,46]. Bone marrow adipose tissue provides a dormancy niche for breast cancer metastases[47], possibly providing metabolic fuel in addition to signals that support quiescence[48]. Individuals with obesity often face increased risk: in primary breast, as well as ovarian, and gastric cancers, excess weight is an established risk factor[7,49]. It is possible that direct tumor-adipocyte signaling contributes to these phenotypes. Given evidence that TNBC both oxidize fatty acids for growth[2] and directly transfer lipolytic cAMP to the tumor niche, systemic lipolysis—whether naturally occurring or induced, as in the case of GLP-1 agonist-mediated weight loss[50,51]—could supplant juxtacrine lipolytic signaling to increase availability of free fatty acids (FFA). In mice bearing *GJB3*[Med] TNBC xenografts, we used a known lipolysis-inducing agent CL316243, which had previously been validated in C57BL/6 male mice to increase circulating FFA. While we did not directly assess its impact on circulating FFA levels in our model, we found CL316243 to increase tumorigenesis and note that lipolysis-inducing agents could feed tumorigenesis in MYC-driven breast cancers with low GJB3. Additional studies with an anti-lipolytic agent are warranted to assess whether systemic or localized decreases of FFA levels may inhibit growth of TNBC.

## Methods

### Ethics declaration
All animal care and rodent experiments comply with the University of California, San Francisco's regulations for controlled substance usage, for biological agent usage, and for animal studies (Institutional Animal Care and Use Committee, Protocol AN200579-00I). All human specimen usage was reviewed and approved by the respective institutional review boards and informed consent was received from participants. The prospective diagnostic 3CB imaging clinical study, initially published in Drukker et al.[17], was approved by the respective institutional review boards at the University of California, San Francisco (UCSF) and at the Moffitt Cancer Center, Tampa, Florida, and followed Health Insurance Portability and Accountability Act-compliant protocols. All study participants provided written informed consent and received compensation for any imaging conducted in addition to the standard of care they were receiving. Invasive tumor samples from participants (Fig. 1a–d and Supplementary Fig. 1a-c) were collected as part of this study with patient informed consent, without additional compensation. For normal and invasive tumor samples used for microdissection and mass-spec analysis (Fig. 1f), all patients provided informed consent and were not reimbursed for participation. For invasive breast cancer, IHC staining (Fig. 3a), all patients provided informed consent and were not reimbursed for participation. Normal reduction mammoplasty samples used for human adipocyte-tumor co-culture (Fig. 3b, e, g and Supplementary Fig. 3a-f) were received from the Cooperative Human Tissue Network (CHTN) with the approval of the IRB of UCSF. Tissues were received as deidentified samples, and all donors provided written informed consent. As the CHTN collects remnant samples the patients were not reimbursed for their participation.

### 3CB patient population
Five hundred women with suspicious mammography findings (BIRADS 4 or greater) were recruited and imaged before their biopsies using a 3-compartment decomposition dual-energy mammography protocol (3CB). This was multicenter study with two recruitment sites: University of California, San Francisco, California and Moffitt Cancer Center, Tampa, Florida. All patients received a biopsy of the suspicious area, and breast biopsies were clinically reviewed by the pathologists. A subset of pathology-proven triple-negative ($n = 6$) and receptor-positive ($n = 40$) invasive cancers were selected for this study. All women received both cranio-caudal (CC) and mediolateral-oblique (MLO) views. Exclusion criteria for the study were no prior cancer, biopsies, or breast ipsilateral alterations, and no occult findings.

### 3CB imaging protocol
The 3CB method combines the dual-energy X-ray mammography attenuations and breast thickness map to solve for the three unknowns water, lipid, and protein content[16]. The Hologic Selenia full-field digital mammography system (Hologic, Inc.) was used to image women with 3CB. Two dual-energy mammograms were acquired on each woman's affected breast using a single compression. The first exposure was made under conditions of regular clinical screening mammogram. The second mammogram was acquired at a fixed voltage (39 kVp) and mAs for all participants. A high-energy exposure (39 kVp/Rh filter) was made using an additional 3-mm plate of aluminum in the beam to increase the average energy of the high-energy image. We limited the total dose of this procedure to be approximately 110% of the mean-glandular dose of an average screening mammogram. The images were collected under an investigational review board approval to measure breast composition. The breast thickness map was modeled using the SXA phantom[52]. The thickness validation procedure concluded in a weekly scanning of specially designed quality assurance phantom[53]. The calibration standards and 3CB algorithms are described in full elsewhere[16,54]. The region of interests of lesions and three surrounding rings of 2 mm distance outward from lesion boundary were derived for water, lipid, and protein maps. The median lipid measures of regions of interest within lesions, three rings outside of lesions, differences, and ratios between lesions and rings were generated for both CC and MLO mammograms. Average values of generated variables of two views were used.

### Histological sectioning, hematoxylin and eosin staining, and adipocyte area quantification
Invasive breast carcinomas were obtained from the Pathology Departments of the University of California, San Francisco (San Francisco, CA) and Moffitt Cancer Center (Tampa, FL). The study population included 39 hormone receptor-positive tumors (32 ER+/PR+/HER2-, 2 ER+/PR−/HER2-, 4 ER+/PR+/HER2+, and 1 ER+/PR−/HER2+), 6 triple-negative (ER−/PR−/HER2-) tumors, and 1 ER−/PR−/HER2+ tumor. Thirty-nine tumors were invasive ductal carcinomas and 7 were invasive lobular carcinomas. Tissue was fixed in 10% formalin and embedded in paraffin, and 4-micron sections were cut for hematoxylin and eosin (H&E) and immunohistochemical ER, PR, and HER2 staining, as well as HER2 fluorescence in situ hybridization (FISH) for a subset of tumors. ER, PR, and HER2 were scored according to ASCO/CAP guidelines[55,56]. An H&E-stained slide demonstrating tumor and sufficient (at least 0.5 cm) NAT was chosen from each of 11 tumors with available slides and subjected to whole slide scanning at 400× magnification using an Aperio XT scanner (Leica Biopsystems, Buffalo Grove, IL). Images were visualized using ImageScope software (Leica Biosystems). For each tumor, 4 representative images at 50× magnification (at least 50 adipocytes per image) from R1 and R3 were analyzed using Fiji imaging software with the opensource Adiposoft v1.13 plugin[57].

### cAMP-dependent lipolysis signature
The cAMP-dependent lipolysis gene signature was generated using RNA-seq data of cAMP-treated adipocytes[19]. Differentially expressed

genes were sorted according to their *P* value and the top 100 upregulated genes were chosen for the signature. This signature was then used to calculated enrichment scores using the single-set gene set enrichment analysis (ssGSEA) method[21]. "cAMP 100 signature" enrichment scores were calculated for a dataset containing multiple samples from multiple regions surrounding breast tumors[20]. The dataset includes samples from the tumor itself (*n* = 9), and NAT 1 cm (*n* = 7), 2 cm (*n* = 5), 3 cm (*n* = 3), and 4 cm (*n* = 4) away from the tumor, in addition to healthy normal samples (*n* = 10). The spatial dataset of multiple regions surrounding breast tumors was downloaded from EMBL-EBI ArrayExpress (Accession E-TABM-276). Raw CEL files were downloaded and processed using custom Affymetrix GeneChip Human Genome U133 Plus 2.0 CDF obtained from BrainArray[58]. The processing and normalization were performed using the Robust Multi-array Average (RMA) procedure on Affymetrix microarray data.

### Laser capture microdissection

Breast tumor tissue was sectioned at 6 μm in a Leica CM 1850 Cryostat (Leica Microsystems GmbH). The sections were mounted on uncharged glass slides without the use of embedding media and placed immediately in 70% ethanol for 30 s. Subsequent dehydration was achieved using graded alcohols and xylene treatments as follows: 95% ethanol for 1 min, 100% ethanol for 1 min (times 2), xylene for 2 min and second xylene 3 min. Slides were then dried in a laminar flow hood for 5 min prior to microdissection. Then, sections were laser captured microdissected with PixCell II LCM system (Arcturus Engineering). Approximately 5000 shots using the 30 micron infrared laser beam were utilized to obtain approximately 10,000 cells per dissection. All samples were microdissected in duplicate on sequential sections.

### SDS-PAGE and In-Gel Digestion

All membranes containing the microdissected cells from breast tumor tissue were removed and placed directly into a 1.5 mL Eppendorf tube. Membranes containing the microdissected cells were suspended in 20 μL of SDS sample buffer, reduced with DTT and heated in a 70–80 °C water bath for approximately 10 min. The supernatant was then electrophoresed approximately 2 cm into a 10% Bis Tris gel, stained with Colloidal Blue with destaining with water, and the region was excised and subjected to in-gel trypsin digestion using a standard protocol. Briefly, the gel regions were excised and washed with 100 mM ammonium bicarbonate for 15 min. The liquid was discarded and replaced with fresh 100 mM ammonium bicarbonate and the proteins reduced with 5 mM DTT for 20 min at 55 °C. After cooling to room temperature, iodoacetamide was added at 10 mM final concentration and samples were placed in the dark for 20 min at room temperature. The solution was discarded and the gel pieces washed with 50% acetonitrile/50 mM ammonium bicarbonate for 20 min, followed by dehydration with 100% acetonitrile. The liquid was removed and the gel pieces were completely dried, re-swelled with 0.5 μg of modified trypsin (Promega) in 100 mM $NH_4HCO_3$, and digested overnight at 37 °C. Peptides were extracted by three changes of 60% acetonitrile/0.1% TFA, and all extracts were combined and dried *in vacuo*. Samples were reconstituted in 35 μL 0.1% formic acid for LC-MS/MS analysis.

### LC-MS/MS analysis, protein identification, and quantitation

Peptide digests were analyzed on a Thermo LTQ Orbitrap Velos ion trap mass spectrometer equipped with an Eksigent NanoLC 2D pump and AS-1 autosampler as described previously[59]. Peptide sequence identification from MS/MS spectra employed the UniProt human protein sequence database, release 2025_02, and database search with SequestHT with Proteome Discoverer Software version 2.5 (Thermo-Fisher Scientific). The dataset contained 1,698,073 MS/MS spectra, with 439,489 peptide-spectrum matches (FDR 0.004) corresponding to 14,501 peptide groups FDR 0.01) and 2751 proteins (FDR 0.01). HNF4α was identified by peptide-spectrum match to MS/MS spectra of the 2+ and 3+ precursors corresponding to the fully tryptic sequence LLPGAVATIVKPLSAIPQPTITK, which was identified with a q value of 0.000. Label-free quantitation was performed using precursor ion intensity. Peptide groups were normalized to the total peptide amount, with the normalization factor being the sum of the sample and the maximum sum of all files analyzed. The peptide group abundances were summed to determine the protein abundance. Protein abundances were scaled so that the averages of all samples were 100.

### Orthotopic xenograft and allograft studies

The human samples used to generate patient-derived xenograft (PDX) tumors, as well as the human non-tumor samples, were previously described[26]. The generation of the MTB-TOM tumor model has been previously described[27]. Four-week-old female WT FVB/N mice (Taconic FVB-F) and immunocompromised NOD/SCID-gamma (NSG) mice (Taconic NODSC) were purchased from Taconic Biosciences. Viably frozen MTB-TOM, HCI002, HCI009, and HCI010 tumor samples were transplanted into the 4th mammary fat pad, following clearance of associated lymph node and epithelium, of respective FVB/N (MTB-TOM) and NSG mice under 2% isoflurane. FVB/N mice were administered dietary doxycycline starting one day before transplant surgery (Bio-Serv #S3888). Tumor growth was monitored daily by caliper measurement in two dimensions. When tumors reached 1 cm (MTB-TOM) or 2 cm (PDX) in any dimension mice were euthanized and tumor and NAT were isolated, and flash-frozen in liquid nitrogen. For the HCC1143 and HS578T control and Cx31 partial expression loss orthotopic xenografts, for the BT549 shRNA knockdown orthotopic xenografts, and for the HCC1143 partial expression loss CL316243 studies, $5 \times 10^5$ cells were resuspended 1:1 with Matrigel (Corning) and injected into the cleared mammary fat pads of 4-week-old WT NSG female mice under 2% isoflurane. Tumor incidence and growth were monitored daily via palpation and caliper measurement, respectively. Mice were euthanized after 180 days or after tumors reached 2 cm in any dimension. For HCC1143 *GJB3^WT^* and *GJB3^Med^* xenografts, a central slice of tumor and surrounding NAT was fixed in 4% paraformaldehyde and embedded in paraffin for histological sectioning, H&E staining, and adipocyte area quantification, while the remaining tumor and NAT tissues were flash-frozen in liquid nitrogen. For other xenografts, NAT was isolated and flash-frozen in liquid nitrogen. For the CL316243 experiment, mice were randomized into experimental groups and moved into new cages immediately post-orthotopic xenograft. The following day, drug treatment was initiated, and mice received vehicle or 1 mg/kg CL316243, delivered by intraperitoneal injection, daily until tumor incidence was recorded via palpation. For the Cx31 shRNA knockdown experiments, mice were randomized into experimental groups (with or without doxycycline). In the shCx31 or shGFP knockdown groups mice were administered doxycycline dietarily starting one day before transplant surgery (Bio-Serv #S3888), while mice in other groups received standard chow. All mice imported from Taconic Biosciences were given 3–7 days to acclimate to our facilities prior to tumor transplantation or cell injection. Mice were moved into new cages immediately following tumor transplantation or cell injection. All mice were maintained at UCSF rodent barrier facilities, on a 12:12 h light:dark cycle, at 68–76 F and 30–70% humidity. Because the vast majority of breast cancers occur in women, female mice were chosen for all orthotopic xenograft and allograft studies. In accordance with the UCSF Institutional Animal Care & Use Committee (IACUC) guidelines, the maximal tumor size permitted was 2 cm in any dimension, and this limit was not exceeded in any of the orthotopic xenograft or allograft studies. All animal study protocols described in this, and other sections were given ethical approval by the UCSF IACUC.

## Immunoblot analysis

Proteins were extracted using RIPA buffer (Thermo) and proteinase (Roche) plus phosphatase (Roche) inhibitor cocktails. Protein extracts were resolved using 4–12% SDS-PAGE gels (Life Technologies) and transferred to nitrocellulose membranes (Life Technologies). Membranes were probed with primary antibodies overnight on a 4 °C shaker, then incubated with horseradish peroxidase (HRP)-conjugated secondary antibodies, and signals were visualized with ECL (Bio-Rad). Primary antibodies targeting the following proteins were used: β-actin (actin) (sc-47778 HRP, Santa Cruz, 1:10,000), pHSL S563 (4139, Cell Signaling, 1:1000), HSL (4107, Cell Signaling, 1:1000), HNF4α (ab41898, Abcam, 1:1000), and Cx31 (ab236620, Abcam, 1:1000). Chemiluminescent signals were acquired with the Bio-Rad ChemiDoc XRS+ System equipped with a supersensitive CCD camera. Where indicated, unsaturated band intensities were quantified using Bio-Rad Image Lab software.

## Cell culture and virus production

A panel of established TN and RP human breast cancer cell lines, and their culture conditions, have previously been described[60]. These cell lines HCC1428 (HCC1428–CRL-2327), T47D (T-47D–HTB-133), HCC3153 (CVCL_3377), HS578T (Hs 578T–HTB-126), BT549 (BT-549–HTB-122) and HCC1143 (HCC1143–CRL-2321) were derived from the primary breast cancer cells of female patients and were originally obtained from the collection of Dr. Adi Gazdar at UT Southwestern Medical Center, or from ATCC. No cell line used in this paper is listed in the database of commonly misidentified cell lines that is maintained by the International Cell Line Authentication Committee (ICLAC) (http://iclac.org/databases/cross-contaminations/). All lines were found to be negative for mycoplasma contamination. Lentiviruses for Cas9 and sgRNAs were produced in 293T cells using standard polyethylenimine (Polysciences Inc.) transfection protocols.

## Dye transfer and FACS analysis

For cancer cell-cancer cell transfer, monolayers of indicated lines (donors) were labelled with 1 μM CalceinAM dye (Life Technologies) at 37 °C for 40 min. Dye-loaded "donor" cells were washed three times with PBS, and then single-cell suspensions of $1.5 \times 10^5$ mCherry-labelled cells (recipients) were added for 5 h. For CBX treatment studies, monolayers of indicated lines (recipients) were pre-treated for 24 h with 150 μM CBX or vehicle. Indicated "donor" cells were loaded in suspension with CalceinAM dye (Life Technologies) at 37 °C for 40 min, washed three times with PBS, and added onto indicated "recipient" cells for 5 h. Dye transfer was quantified by BD LSRFORTESSA or BD LSR II (BD Biosciences). Gating strategy to identify mCherry-positive, Calcein-positive cell population is described in Supplementary Fig. 4. For cancer cell-adipocyte transfer, monolayers of indicated control or Cx31 partial knockout lines (donors) were labelled with 1 μM CalceinAM dye at 37 °C for 40 min. Dye-loaded cells were washed three times with PBS, and then primary mammary adipose tissues (recipient) were added for 5 h. Primary adipose tissue was isolated from co-culture, washed with PBS, and dye transfer was quantified by measurement of total adipose fluorescence using a Tecan fluorescent plate reader.

## Gene expression analysis

TCGA breast-invasive carcinoma dataset was sourced from data generated by TCGA Research Network (https://www.cancer.gov/ccg/research/genome-sequencing/tcga), made available on the University of California, Santa Cruz (UCSC) Cancer Browser. DGE analysis of TN compared to RP patient tumors was calculated using the 'limma' R package[61]. Single-cell RNA-seq data was sourced from data generated by Chung et al.[38]. For the MTB-TOM RNA-seq dataset, MTB-TOM mice (MMTV-rtTA/TetO-MYC)[27] were given doxycycline (n = 10) or standard chow (n = 3), and spontaneous tumors (n = 10) at ethical endpoint

(1 cm in any direction) as well as mammary glands from naïve mice (n = 3) were flash-frozen in liquid nitrogen. Library preparation and Illumina RNA-seq was performed by Q²Solutions (www.q2labsolutions.com). DGE analysis of MTB-TOM compared to normal mammary gland was performed using the DESeq2 package[62]. All RNA was isolated using the RNAeasy kit (Qiagen). See "Data availability" for gene expression analysis datasets.

## ATP quantification

To determine the effects of CBX treatment on ATP levels, tumor cells were seeded in 96-well plates at 5000–7000 cells per well and cultured in the presence of vehicle or 150 μM CBX (Sigma) for 24 h, with triplicate samples for each condition. Relative ATP concentrations were determined using the CellTiter-Glo Luminescent Cell Viability Assay (Promega).

## Isolation of primary mammary adipose tissue

Reduction mammoplasty samples were obtained from the CHTN. Samples were washed in DPBS supplemented with 1% Penicillin/Streptomycin and 0.1% Gentamicin (all GIBCO). Mammary adipose tissue was separated mechanically from epithelial tissue using a razor blade, and was then cryopreserved in freezing medium (10% DMSO (Sigma) in FBS (X&Y Cell Culture)). Normal.

## Immunofluorescence staining and microscopy

For adipose tissue cancer cell co-cultures imaged whole mount, $2 \times 10^6$ of the indicated GFP-labelled cell line was suspended in 500 μL DMEM/F-12(Gibco 11320033) containing 10% FBS and injected into primary mammary adipose tissue from a healthy individual, then cultured at 37 °C for 24 h. For immunofluorescence labeling of co-culture tissues, samples were washed three times in PBS and fixed in 4% paraformaldehyde, permeabilized in 0.5% Triton X-100 for 15 min, and blocked in 10% goat serum in PBS with 0.25 g/L BSA, 0.2% Triton X-100, and 0.41% Tween-20 overnight. Samples were then incubated overnight with primary antibodies (Cx31, WH0002707M1, Sigma, 1:100, and pHSL(S563), 4139, Cell Sig, 1:100), and then overnight with Alexa Fluor-647 or -546 conjugated antibodies. Phalloidin-stained co-cultures were subsequently incubated overnight with Phalloidin-405 (A30104, Invitrogen, 1:200). Finally, using an established protocol for whole mount breast tissue imaging[41], co-culture tissues were cleared through overnight incubation at 4 °C in a "FUnGI" solution of 50% glycerol (vol/vol), 2.5 M fructose, 2.5 M urea, 10.6 mM Tris Base, and 1 mM EDTA. Confocal images were acquired using a Zeiss LSM900 with Airyscan 2 detector. For pHSL(S563) image quantification, fluorescence was measured using Fiji imaging software, and Difference of Gaussians was used for analysis of puncta number and percent area in Fiji. Version 2.10.0. For sectioned adipose tissue co-culture, $1 \times 10^6$ of the indicated mCherry-labelled cell line was injected into primary mammary adipose tissue and cultured at 37 °C for 18 h. The co-cultures were examined using fluorescent microscopy to identify regions of adipose tissue containing mCherry-positive cancer cells. These regions were isolated and fixed in 4% paraformaldehyde and embedded in paraffin. Primary TNBCs used for immunofluorescence were identified and retrieved from the clinical archives of the UCSF Department of Pathology. All tumors consisted of estrogen receptor (ER)-, progesterone receptor (PR)-, and HER2-negative invasive ductal carcinomas. Breast tissue was fixed in 10% formalin and embedded in paraffin. Tumor blocks with sufficient tumor and adjacent (at least 0.5 cm) normal tissue were selected, and 4 μm sections were cut on plus-charged slides for immunofluorescence. Patients provided written informed consent and did not receive reimbursement. This study was approved by the UCSF institutional review board. For immunofluorescence labeling of sectioned co-cultures and primary TNBC, slides were dewaxed in xylene followed by rehydration in graded ethanol (100, 95, 70%) and deionized $H_2O$. Antigen retrieval was

performed in 10 mM Tris, 1 mM EDTA, 0.05% Tween 20, pH 9 at 121 °C for 4 min. Subsequently, tissue sections were blocked in 1% bovine serum albumin and 2% fetal bovine serum in PBS for 5 min, then incubated with primary antibodies (Cx31, 12880, Proteintech, 1:50 and pan-cytokeratin, sc-81714, Santa Cruz, 1:50) overnight at 4 °C. Following several PBS washes, sections were incubated with Alexa Fluor-488 or -568 conjugated antibodies, counterstained with DAPI (Sigma), and mounted using Vectashield (Vector). Epifluorescence images were acquired either by spinning disk microscopy on a customized microscope setup as previously described[63-65] except that the system was upgraded with a next generation scientific CCD camera (cMyo, 293 Photometrics) with 4.5 μm pixels allowing optimal spatial sampling using a Å-60 NA 1.49 objective (CFI 294 APO TIRF; Nikon), or at the UCSF Nikon Imaging Center using a Nikon Ti Microscope equipped with an Andor Zyla 5.5 megapixel sCMOS camera and Lumencor Spectra-X 6-channel LED illuminator. Images were collected using a Plan Apo λ 20×/0.75 lens.

### Generation of Cx31 partial expression loss lines
LentiCas9-Blast (Addgene plasmid #52962) and lentiGuide-Puro (Addgene plasmid #52963) were gifts from Feng Zhang. sgRNAs against Cx31 were constructed using the Feng Zhang Lab CRISPR Design Tool (crispr.mit.edu). sgRNAs used were as follows:

Cx31 exon 1 sg1: CCAGATGCGCCCGAACGCTGTGG (HS578T GJB3[Med-1] and HCC1143 GJB3[Med])

Cx31 exon 1 sg2: CCGGGTGCTGGTATACGTGGTGG (HS578T GJB3[Med-2] and HCC1143 GJB3[Low])

ShRNAs against Cx31 and GFP control were constructed using Tet-pLKO-Puro (Addgene plasmid #21915). shRNAs used were as follows:

shCx31:

shCx31_F: ccggAAGCTCATCATTGAGTTCCTCctcgagGAGGAACTCAATGATGAGCTTttttttg

shCx31_R: aattcaaaaaAAGCTCATCATTGAGTTCCTCctcgagGAGGAACTCAATGATGAGCTT

shGFP[66]:

shGFP_F: CCGGTACAACAGCCACAACGTCTATCTCGACATAGACGTTGTGGCTGTTGTATTTTTG

shGFP_R:CAAAAATACAACAGCCACAACGTCTATGTCGAGATAGACGTTGTGGCTGTTGTACCGG

Lentiviral transduction was performed in DMEM supplemented with 10% FBS and polybrene 10 μg/mL. For sgRNA transduction, Cas9-expressing cells were enriched by Blasticidin (10-15 μg/mL Gemini BioProducts) selection for seven days. Cas9+ cells were subsequently transduced with lentiGuide-Puro (with sgRNAs targeting Cx31), followed by puromycin (1 μg/mL; Gibco) for seven days. Thereafter, clonal selection was performed and clones screened for loss of target gene protein expression by immunoblot analysis. For shRNAs, cells were transduced with Tet-pLKO-Puro (with shRNAs targeting Cx31 or GFP control[66]) followed by puromycin (2 ug/mL; Gibco) for seven days, after which knockdown of target protein was confirmed by immunoblot analysis.

### cAMP quantification
For in vitro studies, tumor cells were seeded in 96-well plates at 5000-7000 cells per well and cultured in the presence of vehicle or 150 μM CBX (Sigma) for 24 h, with triplicate samples for each condition. Changes in cAMP concentration were determined using the cAMP-Glo Assay (Promega).

For in vivo studies, frozen tissue was homogenized using a TissueLyser in 300 μl of 40:40:20 acetonitrile:methanol:water with the addition of 1 nM (final concentration) of D3-[15 N]serine as an internal extraction standard (Cambridge Isotopes Laboratories Inc, DNLM-6863). 10 μl of cleared supernatant (via centrifugation at 20,000 × g., 10 min, at 4 °C) was used for SRM–LC-MS/MS using a normal-phase Luna NH2 column (Phenomenex). Mobile phases were buffer A (composed of 100% acetonitrile) and buffer B (composed of 95:5 water:acetonitrile). Solvent modifiers were 0.2% ammonium hydroxide with 50 mM ammonium acetate for negative ionization mode. cAMP levels were analyzed using the MassHunter software package (Agilent Technologies) by quantifying the transition from parent precursor mass to product ions.

### cAMP transfer
For cancer cell-adipocyte transfer, monolayers of indicated control or Cx31 partial knockout lines (donors) were labelled with 2 μM fluo-cAMP (Biolog Life Science Institute) at 37 °C for 30 min. cAMP-loaded cells were washed three times with PBS, and then primary mammary adipose tissues (recipient) were added for 5 h. Primary adipose tissue was isolated from co-culture, washed with PBS, and cAMP transfer was quantified by measurement of total adipose fluorescence using a Tecan fluorescent plate reader.

### Preadipocyte differentiation and qRT-PCR
Primary mouse preadipocytes were differentiated as previously described[67]. Monolayers of differentiated adipocytes were washed with PBS, and then treated with vehicle or 10 μM forskolin (Sigma), or seeded with $1 \times 10^5$ of the indicated cancer lines. Total RNA was isolated from co-cultures after 20 h using the RNeasy kit (Qiagen). One μg of total RNA was reverse transcribed using iScript cDNA synthesis kit (Bio-Rad). The relative expression levels of *UCP1*, *FABP4*, and *GAPDH* were analyzed using a SYBR Green Real-Time PCR kit (Thermo) with an Applied Biosystems QuantStudio 6 Flex Real-Time PCR System thermocycler (Thermo). Variation was determined using the ΔΔCT method[68](48) with *GAPDH* mRNA levels as an internal control. Mouse-specific primers used were as follows:

GAPDH forward: CCAGCTACTCGCGGCTTTA

GAPDH reverse: GTTCACACCGACCTTCACCA

UCP1 forward: CACCTTCCCGCTGGACACT

UCP1 reverse: CCCTAGGACACCTTTATACCTAATGG

FABP4 forward: ACACCGAGATTTCCTTCAAACTG

FABP4 reverse: CCATCTAGGGTTATGATGCTCTTCA

### Proliferation and viability assays
To determine the effects of Cx31 partial knockout on cell proliferation and viability, the indicated cell lines were seeded in 6-well plates at $1.5 \times 10^5$ cells/well. Cells were harvested at 24, 48, and 72 h. Cell counts and cell viability by trypan blue exclusion were determined using the Countess Automated Cell Counter (Life Technologies) according to the manufacturer's instructions.

### Statistics & reproducibility
Prism software (v 10.4.1) was used to generate and analyze Spearman correlation (Fig. 1d) and the survival plots (Fig. 4b, c, and g). Survival plot $P$-values were generated using a log-rank test. Correlation $P$ values were generated using ordinary one-way ANOVA with multiple comparisons (Figs. 1f, h, 2b, 3f, h, 4a right, 4d and f), two-way ANOVA with multiple comparisons (Fig. 1c), repeated measures one-way ANOVA with multiple comparisons (Figs. 1b and 3e and g), repeated measures mixed effects model with multiple comparisons (Fig. 1e), and unpaired two-tailed $t$ test (Figs. 2a and c, 4a left and center, 4e, and Supplementary Fig. 1a, b). These analyses were performed using PRISM software. DGE analysis of TN compared to RP patient tumors (Fig. 2d) was calculated using the "limma" R package[61]. DGE analysis of MTB-TOM compared to normal mammary gland (Fig. 2f) was performed using the DESeq2 package[62]. Differential expression analyses (Fig. 2d and f) were calculated with a false discovery rate of 0.05. Biological replicates are shown in Figs. 1b–e, g, h, 2a–c, 3e–h, 4a, c–f, and Supplementary Fig. 1a, b. Technical duplicates for each biological replicate are shown in Fig. 1f. Immunofluorescence images shown in Fig. 3a, b and d and Supplementary Fig. 3a, b and e are representative from experiments repeated in 3 independent biological replicates.

No statistical method was used to predetermine sample size. The investigators were not blinded to allocation during in vivo experiments. The investigators were blinded during immunofluorescence analysis. All in vivo studies were randomized as tumors reached predetermined volume, or palpability, on a per-experiment basis as described in *Orthotopic xenograft and allograft studies* above. No samples from in vivo studies were processed and then excluded from immunoblot, intracellular cAMP or Adiposoft analysis, except where tumor size was insufficient to allow for analysis via multiple methods. No mice that completed the xenograft or allograft studies were excluded from analyses. For all in vitro and in vivo experiments, sample size was not chosen with consideration of adequate power to detect a pre-specified effect size. For in vitro studies, all completed experiments are reported. For in vivo studies, number of indicated mice per individual experiment represents the total number of mice treated and analyzed.

### Reporting summary

Further information on research design is available in the Nature Portfolio Reporting Summary linked to this article.

## Data availability

The dataset used in Fig. 1e containing multiple samples from multiple regions surrounding breast tumors was previously published[20] and is available in the ArrayExpress database[69] under accession code E-TABM-276. Data used to generate the cAMP 100 signature applied in Fig. 1e was previously published[19] and are available in the ArrayExpress database[69] under accession code E-MTAB-2602. Protein MS data containing HNF4α spectral counts from healthy patient normal breast, stroma and breast lesion LCM generated for Fig. 1f and Supplementary Data 2a, b are available through Proteome Xchange under accession code MSV000097890. The dataset used in Fig. 2d is available from The Cancer Genome Atlas Program (TCGA) database under accession code phs000178 [https://www.cancer.gov/ccg/research/genome-sequencing/tcga]. Our annotations of the TCGA dataset used in Fig. 2d are available [https://bitbucket.org/jeevb/brca]. The single-cell RNA-seq dataset for Fig. 2e and Supplementary Fig. 2 was previously published[38] and is available in the NCBI Gene Expression Omnibus database under accession code GSE75688. The RNA-seq dataset used in Fig. 2f and Supplementary Data 3 in the MTB-TOM (MMTV-rtTA/TetO-MYC) model comparing spontaneous tumor tissue, from mice on dietary doxycycline, to control non-tumor tissue, from mice on standard chow, was previously published[70] and is available in the NCBI Gene Expression Omnibus under GSE130921. Source data are provided with this paper.

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

## Acknowledgements

This work was supported, in part, by the US Department of Defense–Congressionally Directed Medical Research Programs' Era of

Hope Scholar award W81XWH-12-1-0272 and W81XWH-21-1-0774 (A.G.), the US National Institutes of Health (NIH) grants R01CA17447 and 2R01EB028148 (A.G.) and R01 CA056721 (Z.W.), the Atwater Foundation and Bechtle Family Foundation (A.G.), the Breast Cancer Research Fund and Subramanian Breast Cancer Support Fund (H.R., A.G.), the EMBO postdoctoral fellowship ALTF 159-2017 (Ju.W.), the US K99/R00 NIH Pathway to Independence Award DK110426 (K.S.), the US National Cancer Institute grants R01CA166945 and R01CA257652 (J.A.S.), and the US NIH F99/K00 Predoctoral to Postdoctoral Transition Award F99CA212488 (R.C.) and the US NIH F31 Predoctoral Individual National Research Service Award F31CA243468 (J.W.). Tissue samples were provided by the Cooperative Human Tissue Network (RRID: SCR_004446), a National Cancer Institute supported resource. Other investigators may have received samples from these same tissue specimens. The authors thank A. Welm for guidance in the use of patient-derived xenografts, A. Tward for technical guidance and helpful discussions, K.A. Fontaine for helpful discussions and comments on the manuscript, and S. Samson for a helpful consumer advocate perspective and guidance on our work.

## Author contributions

R.C. and A.G. conceived of research. J.W. and R.C. designed and contributed to all in vitro and ex vivo studies, and to in vivo mouse studies, contributed to all data analysis, and wrote the manuscript. Se.M. analyzed 3CB data. L.J.Z. designed LCM and conducted proteomic analyses. Su.M. conducted the LCM. D.A. conducted lipolysis signature enrichment and scRNA-seq analyses. A.B. contributed to immunofluorescent staining and microscopy, and generation of Cx31 partial knockout lines. D.V. characterized Cx31 partial depletion lines, generated Cx31 inducible short hairpin lines, contributed to Cx31 knockdown and CL316243 mouse studies and provided helpful discussions. S.K.A. contributed to generation of Cx31 inducible short hairpin lines. R.N. contributed to coculture studies, immunofluorescent staining, tissue clearing, and microscopy, and provided valuable discussion. Y.C. conducted preadipocyte differentiation. C.B. and S.L. conducted mass spectrometry for cAMP. C.M. generated Cas9-expressing cancer cell lines. Ju.W. and E.W. isolated primary mammary adipose tissue. E.J.H. contributed to coculture microscopy and provided valuable discussion. J.D.G. and D.S. conducted FACS analysis. K.S. provided cAMP-dependent lipolysis signature. M.G. contributed to RNA-seq analysis. H.N. contributed to coculture studies. K.M.A. supervised FACS analysis. Z.W. supervised primary mammary adipose tissue isolation and provided valuable discussion. D.K.N. supervised mass spectrometry for cAMP and provided valuable discussion. S.K. supervised preadipocyte and cAMP-dependent lipolysis studies and provided valuable discussion. A.J.B. supervised enrichment analysis. M.E.S. and D.C.L. designed and supervised LCM and proteomics. H.R. provided valuable discussion. G.K. designed and conducted histological analyses and provided valuable discussion. J.A.S. conceived of applying 3CB study data, supervised the 3CB study and data analysis, and provided valuable discussion. A.G. supervised the study, and provided valuable discussion and intellectual input. All authors edited the manuscript.

## Competing interests

The authors declare no competing interests.

## Additional information

[1]Department of Cell & Tissue Biology, University of California, San Francisco, San Francisco, CA, USA. [2]Biomedical Sciences Graduate Program, University of California, San Francisco, San Francisco, CA, USA. [3]Department of Radiology & Biomedical Imaging, University of California, San Francisco, San Francisco, CA, USA. [4]Department of Biochemistry, Vanderbilt University School of Medicine, Nashville, TN, USA. [5]Jim Ayers Institute for Precancer Detection and Diagnosis, Vanderbilt-Ingram Cancer Center, Nashville, TN, USA. [6]Department of Pathology, Vanderbilt University School of Medicine, Nashville, TN, USA. [7]Faculty of Biology, Technion, Israel Institute of Technology, Haifa, Israel. [8]The Taub Faculty of Computer Science, Technion, Israel Institute of Technology, Haifa, Israel. [9]Department of Medicine, University of California, San Francisco, San Francisco, CA, USA. [10]Diabetes Center, University of California, San Francisco, San Francisco, CA, USA. [11]Eli and Edythe Broad Center of Regeneration Medicine and Stem Cell Research, University of California, San Francisco, San Francisco, CA, USA. [12]Tongji Medical College, Huazhong University of Science and Technology, Wuhan, China. [13]Department of Chemistry, University of California, Berkeley, Berkeley, CA, USA. [14]Department of Molecular & Cell Biology, University of California, Berkeley, Berkeley, CA, USA. [15]Department of Nutritional Sciences & Toxicology, University of California, Berkeley, Berkeley, CA, USA. [16]Department of Anatomy, University of California, San Francisco, San Francisco, CA, USA. [17]Helen Diller Family Comprehensive Cancer Center, University of California, San Francisco, San Francisco, CA, USA. [18]Center for Cancer Research, Medical University of Vienna, Vienna, Austria. [19]Department of Microbiology & Immunology, University of California, San Francisco, San Francisco, CA, USA. [20]Sandler Asthma Basic Research Center, University of California, San Francisco, San Francisco, CA, USA. [21]Department of Medicine and Molecular Pharmacology, Albert Einstein College of Medicine, Bronx, NY, USA. [22]Division of Endocrinology, Diabetes and Metabolism, Beth Israel Deaconess Medical

Center, Harvard Medical School, Boston, MA, USA. [23]Howard Hughes Medical Institute, Chevy Chase, MD, USA. [24]Department of Medical Oncology & Therapeutics Research, City of Hope Comprehensive Cancer Center, Duarte, CA, USA. [25]Department of Pathology, University of California, San Francisco, San Francisco, CA, USA. [26]Cancer Center, University of Hawaii, Honolulu, HI, USA. [27]These authors contributed equally: Jeremy Williams, Roman Camarda. ✉e-mail: Andrei.Goga@ucsf.edu

