## [Transparent Peer Review file · Nature Communications]

Tumor cell-adipocyte gap junctions activate lipolysis and contribute to breast tumorigenesis

Corresponding Author: Dr Andrei Goga

Version 0:

Reviewer comments:

Reviewer #2

(Remarks to the Author)

The updated manuscript from the Goga lab and their collaborators describes the outcomes from a comprehensive series of experiments to implicate adipocyte-cancer cell gap junctions are essential for tumorigenesis, especially Cx31 in triple-negative breast cancer.

I appreciate the nature in which my comments have been addressed and thank the authors for their comprehensive rebuttal.

Original comments:

Points 1 and 16 (existing literature) – I appreciate the 50 citation limit and can see an appropriate acknowledgement of the existing and relevant literature in the revised manuscript.

Point 2 (pHSL/HSL) – the ambiguity in the western blot quantification has been resolved.

Point 3 & 5 (Fig 3F and statistics) – this has been resolved. Importantly, the appropriate use of statistical tests has been reported.

Point 4 (quality of presentation) – I suggest that Figure 2 could benefit from a more careful choice of colour. For example, panels B, D, and F share the same colours, but I do not think that their significance in panel B is the same as that of D and F. Please revise.

Point 6 – (number of observations) – the details regarding the number of observations have significantly been improved.

Thank you.

Point 7 (conclusions) – the authors have added significant new data that resolves my concerns regarding sufficient evidence to support their conclusions.

Point 8 (lipolysis and CM/co-culture) – I appreciate the view of the authors on the various settings where adipocyte lipolysis is stimulated. I think it would be more balanced if the authors acknowledged that cancer cells can stimulate adipocyte lipolysis via other mechanisms alongside their conclusion at the end of the paragraph starting with "We next sought..." on page 11 (the lack of line numbers makes it hard to easily direct the authors to the relevant section of the manuscript)

Point 9 (NT vs adipocyte-cancer) – the inclusion of additional experimental data has addressed this major limitation. That said, the results presented in Figure 4F need to be reconsidered as I believe that the pooling of adipocyte size data (i.e. individual adipocytes) from 5 or 4 mice is not appropriate, and the data should be collapsed into a mean per mouse and 4 or 5 dots reported. The very low p values are simply a consequence of 100's-1000's data points in each group.

Point 10 (cAMP, gap junctions) – the additional experimental data addresses this concern.

Point 11 (in vivo treatment with CL316243) – I still believe that the lack of evidence that CL316243 stimulated lipolysis in female NOD-SCID mice is a major limitation. Relying on evidence from male C57BL6 mice (ref #22) is limiting, given that CL316243 acutely reduces plasma fatty acids but also reduces circulating glucose levels and increases insulin. I am not suggesting going back to repeat the study to measure body composition (fat mass) or blood/plasma free fatty acid levels, so raising this as a limitation of the study should suffice.

Points 12 (activated lipolysis) and 13 (HS578T tumorigenesis) – the new data addresses these concerns.

Point 14 (BC subtypes) – the additional information in Table S4 assists the reader in appreciating the subtype composition of these data used in the relevant analyses of clinical data.

Point 15 (ATP and viability) – the inclusion of trypan blue exclusion complements the previous measures.

Point 17 (clarity and context) – the revised manuscript has improved the clarity of the specific new insights the authors are reporting. That said, there are some areas of opportunity to acknowledge the existing mechanisms and how the new insights extend and complement the prevailing literature.

Additional Minor Issues:

1. MTB-TOM and MTBTOM are both used.
2. I think it is appropriate that the authors make sure that the conclusions made are limited to TNBC as it is conceivable that other proteins participate in gap-junction formation in RP and other BC subtypes.
3. I think, and I could be wrong, that the “field” has settled on FAPT4 rather than aP2.
4. Page 6 – I suggest splitting or inserting a reference after “...in cancer” in the following sentence to make clear the supporting evidence for the statement of fact - “Connexins were long thought to play tumor-suppressive roles in cancer, but recent evidence from a variety of tumor types has challenged this notion [28-31].”

Reviewer 4 comments

I have been asked by the Editor to also review the rebuttal to Reviewer 4's comments.

Original comments:

Point 1 – I think the reviewers have adequately addressed this comment, which can be argued to fall slightly outside the primary scope of the project. It is conceivable that the Editor may think there is value in including a short “Discussion” paragraph on obesity and BC and these new insights.

Point 2 – the focus on TNBC is appropriate, but even in this revised manuscript, I am not convinced there is sufficient acknowledgement that gap junctions likely play a role in other subtypes of BC, like via other connexin genes or other relevant players. A brief statement on this in an appropriate spot in the manuscript is warranted. Otherwise, a reader may come to the same conclusion as Reviewer 4.

Point 3 – this is an interesting point, but I support the authors' point regarding the technical challenges of testing this.

Point 4 – like the above point, this is an interesting question and, again, technically challenging. It was unclear whether the revised manuscript acknowledges this limitation (Cx31 loss-of-function in adipocytes).

Points 5 and 6 – I again think that these are relevant points that should be discussed in the manuscript. However, the current version is not structured in a classical manner (Intro, Results, Discussion, Methods, etc.), and so the lack of a Discussion sub-section means that these topics (transfer of other factors, mechanisms of oncogenic adipocyte lipolysis) are not covered. I think the manuscript would benefit from this, but also acknowledge that this is up to the Editor.

Point 7 – a statement similar to “we are not proposing evidence that intracellular cAMP levels differ in triple-negative versus receptor-positive breast cancers, but that elevated Cx31 levels permit diffusion of cAMP signal from TNBC tumor into the adipose tumor microenvironment.” acknowledging the limitations of the proposed mechanisms should be added. Likewise, the lack of well-defined mechanisms regulating Cx31 gap junction formation should be acknowledged.

Point 8 – this point has been addressed.

Point 9 – I feel that the authors have addressed this point, but I do think it would be wise to review the revised manuscript to confirm that this rationale to assess homotypic tumour cell gap junctions before heterotypic gap junctions is clearly stated.

Point 10 – this has been addressed.

Quality 1 – this has been adequately addressed.

Quality 2 – I am not an expert in microscopy, but I believe this limitation has been addressed.

Quality 3 - this has been adequately addressed.

Reviewer #5

(Remarks to the Author)

In this resubmission of their manuscript, Carmarda et al. address the role of Cx31 in lipolysis in adipocytes of normal adjacent tissue using a number of techniques and different systems. The work provides some novel insight into the field. Most of the questions/concerns raised in the first round of reviews were addressed to sufficient degree with the following exceptions:

The immunofluorescence co-labeling studies in Fig 3B and D are not compelling as pointed out in the first round of reviews. The cell interfaces between the adipocytes and cell line are not clear – particularly for the HCC1143 cell membranes in B and D. In B upper row the typical Cx staining is everywhere except the interfaces they point out and it seems like the typical Cx31 staining could be only between adipocytes. For D, the Cx31 staining seems to be evenly disbursed in a non-typical manner. Staining with a membrane marker and showing the cell-cell interface in 3D presentation would help.

The other significant issue not fully addressed is the relative importance of Cx31 exchange of cAMP to adipocyte lipolysis. CBX only caused an apparent 1/3 reduction in dye transfer in Fig. 2C and it is unclear from 3E and G whether the reduction of fluorescence observed with reduced GJB3 levels is 20%, 50% or 80% of the wt level – I understand the levels are significantly reduced but not whether it is compared to the extent of the total dye transfer. The point is that there seems to be significant residual intercellular communication that exists that should confound the results. Therefore “Thus, cAMP is transferred from TN breast cancer cells to adipocytes in a Cx31-dependent manner.” is overstated unless you show a zero transfer level upon treatment.

Other minor issues

“Additionally, in 3 of the 4 models examined we found an increase in HNF4a protein or in phospho-HSL/total HSL ratio (Fig. 1, G and H).” – I see the total protein differences but not significant differences in the pHSL ratio.

In Fig 4 why do the medium Cx31 expressing cells have the most pHSL?

Wording issues

“We identify connexin 31 (GJB3), which promotes receptor triple negative breast cancer growth and activation of lipolysis in vivo.” Weird sentence. It implies the authors were the first to identify Cx31.

“We found a marked positive correlation ($R = 0.5818$, $p = 0.0656$) between the change in lipid content and adipocyte area (Fig. 1D).” You found a trend, not a marked correlation.

“In addition, dye uptake in HCC1143 cells was significantly reduced (30.63%, $p < 0.0001$).” Since gap junctions can act as “hemichannels” and uptake dye or other molecules from the media use “transfer” or something else instead.

In Fig 3H the significant difference comparisons are hard to parse particularly in the HCC1143 low column as is has “ns” under a and apparently other comparisons

“Adipocytes were then isolated from the tumor cells and assayed for fluo-cAMP. We found that cAMP transfer occurred from control cells to adipocytes from all three patients (Fig. 3G).” Control isn’t very informative here – I presume they mean GJB3WT.

Reviewer Comments - Final Revisions - Nature Communications

Reviewer #2 (Remarks to the Author):

The updated manuscript from the Goga lab and their collaborators describes the outcomes from a comprehensive series of experiments to implicate adipocyte-cancer cell gap junctions are essential for tumorigenesis, especially Cx31 in triple-negative breast cancer.

I appreciate the nature in which my comments have been addressed and thank the authors for their comprehensive rebuttal.

Reviewer #2: Points 1 and 16 (existing literature) – I appreciate the 50 citation limit and can see an appropriate acknowledgement of the existing and relevant literature in the revised manuscript.

Response: We thank Reviewer 2 for the positive response to our citation of existing work in the revised manuscript.

Reviewer #2: Point 2 (pHSL/HSL) – the ambiguity in the western blot quantification has been resolved.

Response: We appreciate that this issue has been adequately resolved.

Reviewer #2: Point 3 & 5 (Fig 3F and statistics) – this has been resolved. Importantly, the appropriate use of statistical tests has been reported.

Response: We thank Reviewer 2 for their comment. Considerable effort was taken to appropriately use and report statistical tests in the revised manuscript.

Reviewer #2: Point 4 (quality of presentation) – I suggest that Figure 2 could benefit from a more careful choice of colour. For example, panels B, D, and F share the same colours, but I do not think that their significance in panel B is the same as that of D and F. Please revise.

Response: In response to this input, we have revised colors for Figure 2 panels D and F in the resubmission (and pasted below for your review) to reflect their distinct meanings from other colors shown in Figure 2.

Reviewer #2: Point 6 – (number of observations) – the details regarding the number of observations have significantly been improved. Thank you.

Response: We thank Reviewer 2 for their initial remarks and for this comment.

Reviewer #2: Point 7 (conclusions) – the authors have added significant new data that resolves my concerns regarding sufficient evidence to support their conclusions.

Response: We thank Reviewer 2 for their time and help in the initial review and additional comments during the resubmission which have improved the presented data and added clarity.

Reviewer #2: Point 8 (lipolysis and CM/co-culture) – I appreciate the view of the authors on the various settings where adipocyte lipolysis is stimulated. I think it would be more balanced if the authors acknowledged that cancer cells can stimulate adipocyte lipolysis via other mechanisms alongside their conclusion at the end of the paragraph starting with "We next sought..." on page 11 (the lack of line numbers makes it hard to easily direct the authors to the relevant section of the manuscript)

Response: In this revision, we have expanded the *Introduction* to acknowledge other mechanisms by which both cancer cells and adipocytes can remotely confer lipolytic signaling to adipose tissue. We hope that these added acknowledgements address Reviewer 2's suggestion for increased clarity on the known, indirect mechanisms by which cancer cells can stimulate adipocyte lipolysis. We also comment on other mechanism of inducing lipolysis in the second paragraph of the *Discussion*.

Reviewer #2: Point 9 (NT vs adipocyte-cancer) – the inclusion of additional experimental data has addressed this major limitation. That said, the results presented in Figure 4F need to be reconsidered as I believe that the pooling of adipocyte size data (i.e. individual adipocytes) from 5 or 4 mice is not appropriate, and the data should be collapsed into a mean per mouse and 4 or 5 dots reported. The very low p values are simply a consequence of 100's-1000's data points in each group.

Response: We have checked our original dataset for this panel, and we unfortunately only have pooled adipocyte size for each experimental group of mice. We indicated in the revised manuscript that individual adipocyte area was calculated and pooled for each group before statistical comparison was performed. Despite the pooling, the mean adipocyte area was significantly different between the groups as shown in the figure, and in the revised panel we have indicated that $p < 0.05$ based on this input.

Reviewer #2: Point 10 (cAMP, gap junctions) – the additional experimental data addresses this concern.

Response: We thank Reviewer 2 for the opportunity to improve upon the manuscript here.

Reviewer #2: Point 11 (in vivo treatment with CL316243) – I still believe that the lack of evidence that CL316243 stimulated lipolysis in female NOD-SCID mice is a major limitation. Relying on evidence from male C57BL6 mice (ref #22) is limiting, given that CL316243 acutely reduces plasma fatty acids but also reduces circulating glucose levels and increases insulin. I am not suggesting going back to repeat the study to measure body composition (fat mass) or blood/plasma free fatty acid levels, so raising this as a limitation of the study should suffice.

Response: We acknowledge the limitation in our C1316243 study and reliance on this reference. In our updated *Discussion* section, we now include new text acknowledging that “we did not directly assess circulatory FFA levels.” We also propose future studies to directly test anti-lipolytic agents.

Reviewer #2: Points 12 (activated lipolysis) and 13 (HS578T tumourigenesis) – the new data addresses these concerns.

Response: We appreciate that these points have been adequately addressed.

Reviewer #2: Point 14 (BC subtypes) – the additional information in Table S4 assists the reader in appreciating the subtype composition of these data used in the relevant analyses of clinical data.

Response: We thank Reviewer 2 for their positive reception of this addition.

Reviewer #2: Point 15 (ATP and viability) – the inclusion of trypan blue exclusion complements the previous measures.

Response: We thank Reviewer 2 for their response to our rebuttal on this point and to the trypan blue exclusion data we subsequently added into the figure.

Reviewer #2: Point 17 (clarity and context) – the revised manuscript has improved the clarity of the specific new insights the authors are reporting. That said, there are some areas of opportunity to acknowledge the existing mechanisms and how the new insights extend and complement the prevailing literature.

Response: We have now included and expanded **Introduction** and **Discussion** sections, since the prior manuscript was prepared for a more 'minimal' Nature submission. We feel including these sections have allowed us to address and extend the prior literature.

Additional Minor Issues:

Reviewer #2: 1. MTB-TOM and MTBTOM are both used.

Response: We thank Reviewer 2 for bringing this to our attention. We changed any uses of “MTBTOM” to the correct “MTB-TOM” in each instance.

Reviewer #2: 2. I think it is appropriate that the authors make sure that the conclusions made are limited to TNBC as it is conceivable that other proteins participate in gap-junction formation in RP and other BC subtypes.

Response: We take this to mean, that the regulatory mechanisms governing GJ formation and function may differ between TNBC and other BC subtypes, and we certainly agree that this could be the case. In the expanded **Discussion** section of this resubmitted manuscript we address limitations of the study more fully and point out both in the Summary, Introduction and Discussion that the reliance of fatty acid oxidation and greatest expression of Cx31 is observed in the more aggressive MYC^{High} TNBC subset.

Reviewer #2: 3. I think, and I could be wrong, that the “field” has settled on FAPT4 rather than aP2.

Response: We take Reviewer 2 in this case to mean that the name **FABP4** is more commonly recognized than aP2. We have taken this input and appropriately adjusted the name used in this revised submission.

Reviewer #2: 4. Page 6 – I suggest splitting or inserting a reference after “...in cancer” in the following sentence to make clear the supporting evidence for the statement of fact - “Connexins were long thought to play tumor-suppressive roles in cancer, but recent evidence from a variety of tumor types has challenged this notion [28-31].”

Response: We have inserted a reference describing earlier studies which presented tumor-suppressive roles for gap junctions.

Reviewer 2: I have been asked by the Editor to also review the rebuttal to Reviewer 4's comments.

Original comments:

Reviewer #2: Point 1 – I think the reviewers have adequately addressed this comment, which can be argued to fall slightly outside the primary scope of the project. It is conceivable that the Editor may think there is value in including a short “Discussion” paragraph on obesity and BC and these new insights.

Response: We have now included an expanded *Discussion* section in the revised submission which discusses obesity within the context of our findings.

Reviewer #2: Point 2 – the focus on TNBC is appropriate, but even in this revised manuscript, I am not convinced there is sufficient acknowledgement that gap junctions likely play a role in other subtypes of BC, like via other connexin genes or other relevant players. A brief statement on this in an appropriate spot in the manuscript is warranted. Otherwise, a reader may come to the same conclusion as Reviewer 4.

Response: In this revised manuscript we have included a discussion section which we hope addresses this point. While we discuss that the predominant findings were most relevant to the TNBC subtype (**Figure 2D** and below), we also note that: "Several GJB family members are differentially increased when comparing transcript levels in patient TN to RP tumors, suggesting multiple gap junction proteins may be relevant to tumor growth *in vivo*." **Figure 2E** (also pasted below) points out that several classes of gap junction genes are highly expressed in both Triple Negative (TN) and Receptor Positive (RP) breast cancer subtypes.

Reviewer #2: Point 3 – this is an interesting point, but I support the authors' point regarding the technical challenges of testing this.

Response: We agree that understanding impacts of GJB3 perturbation in the context of an existing tumor is interesting and important. We thank Reviewer 2 for their understanding regarding the technical challenges we faced trying to delineate relative contributions of Cx31 to TNBC tumor formation and progression *in vivo*.

Reviewer #2: Point 4 – like the above point, this is an interesting question and, again, technically challenging. It was unclear whether the revised manuscript acknowledges this limitation (Cx31 loss-of-function in adipocytes).

Response: In the revised manuscript we have included an expanded *Discussion* section, which we hope more clearly addresses this point. It includes the text: “The present data do not, however, exclude transfer of other factors by these GJB3 gap junctions, or transfer of cAMP to adipocytes by other kinds of gap junctions.... Fully isolating the contribution of Cx31 to this phenotype presents a technical challenge. The inability to generate

fully Cx31-null patient-derived TNBC cell lines suggests that some degree of expression is required to maintain tumor cell proliferation.”

Figure 4. e Fold change in cAMP levels in HCC1143 GJB3^{Med} xenografts versus HCC1143 GJB3^{WT} xenografts. f Adipocyte area adjacent to HCC1143 GJB3 Med xenografts (pooled from n = 5 tumors, n = 517 adipocytes) and HCC1143 GJB3 WT xenografts (pooled from n = 4, n = 771 adipocytes) and area in control non-tumor (NT) tissue (pooled n=3 mice, n = 2611 adipocytes) ($p < 0.0001$).

Reviewer #2: Points 5 and 6 – I again think that these are relevant points that should be discussed in the manuscript. However, the current version is not structured in a classical manner (Intro, Results, Discussion, Methods, etc.), and so the lack of a Discussion sub-section means that these topics (transfer of other factors, mechanisms of oncogenic adipocyte lipolysis) are not covered. I think the manuscript would benefit from this, but also acknowledge that this is up to the Editor.

Response: In this new draft, we have restructured the paper in a classical manner and included an expanded *Introduction* and *Discussion* section with the goal of addressing these points. We thank Reviewer 2 for the opportunity to better-address these relevant topics.

Reviewer #2: Point 7 – a statement similar to “we are not proposing evidence that intracellular cAMP levels differ in triple-negative versus receptor-positive breast cancers, but that elevated Cx31 levels permit diffusion of cAMP signal from TNBC tumor into the adipose tumor microenvironment.” acknowledging the limitations of the proposed mechanisms should be added. Likewise, the lack of well-defined mechanisms regulating Cx31 gap junction formation should be acknowledged.

Response: In the newly expanded *Discussion* section, we add acknowledgement of these limitations (“The present data also do not indicate that intratumor cAMP levels are greater in TN compared to RP tumors, but that elevated Cx31 in TNBC permits increased transfer of cAMP signal from tumor to adipose tumor microenvironment and increased lipolysis in tumor-adjacent adipocytes,”) which we hope addresses these concerns.

Reviewer #2: Point 8 – this point has been addressed.

Response: We appreciate Reviewer 2’s reception of our response to this point.

Reviewer #2: Point 9 – I feel that the authors have addressed this point, but I do think it would be wise to review the revised manuscript to confirm that this rationale to assess homotypic tumour cell gap junctions before heterotypic gap junctions is clearly stated.

Response: We have amended the text introducing Figure 2A to state, “Using a well-established dye transfer assay, we first probed for presence of functional gap junctions between breast cancer cells. Because gap junction function in breast tumors has not been clearly defined, we tested whether the TNBC cell line HCC1143 or the more indolent RP cell line T47D could transfer gap-junction dependent dyes to the same tumor cell line.” We hope that this appropriately captures our rationale for first presenting these assays.

Reviewer #2: Point 10 – this has been addressed.

Response: We thank Reviewer 2 for evaluating our comments on this topic.

Reviewer #2: Quality 1 – this has been adequately addressed.

Response: We thank Reviewer 2 for reexamining our use of quantification and statistics.

Reviewer #2: Quality 2 – I am not an expert in microscopy, but I believe this limitation has been addressed.

Response: We appreciate that Reviewer 2 accepts our provided rationale on this limitation. In the revised manuscript we have included additional high resolution (63X magnification) microscopy of cocultures at increased magnification (now **Fig. 3D** and included below). These studies further highlight the association of tumor-adipocyte and Cx31 expression at sites of contact. This new figure panel was added since the prior revisions.

Figure 3d. Staining with Cx31 (magenta), pHSL(S563) (yellow), and phalloidin (blue), of healthy patient primary mammary tissue (WD50175) injected with GFP-expressing HCC1143 GJB3^{WT} (top), HCC1143-GJB3^{Low} (middle), or T47D cells (bottom) and co-cultured overnight. White arrowheads indicate Cx31 staining at GFP cancer cell-adipocyte interface. Scale bar, 20 μ m.

Reviewer #2: Quality 3 - this has been adequately addressed.

Response: We agree with Reviewer 2 that these *in vivo* models do independently support the significant impact of GJB3 depletion on tumor-free survival and ethical endpoint.

We thank Reviewer 2 for their insights and help in improving this manuscript. The coauthors greatly appreciate the time taken to provide the initial and subsequent reviews, as well as to respond for comments from another prior Reviewer.

Reviewer #5 (Remarks to the Author):

In this resubmission of their manuscript, Carmarda et al. address the role of Cx31 in lipolysis in adipocytes of normal adjacent tissue using a number of techniques and different systems. The work provides some novel insight into the field. Most of the questions/concerns raised in the first round of reviews were addressed to sufficient degree with the following exceptions:

Reviewer #5: The immunofluorescence co-labeling studies in Fig 3B and D are not compelling as pointed out in the first round of reviews. The cell interfaces between the adipocytes and cell line are not clear – particularly for the HCC1143 cell membranes in B and D. In B upper row the typical Cx staining is everywhere except the interfaces they point out and it seems like the typical Cx31 staining could be only between adipocytes. For D, the Cx31 staining seems to be evenly disbursed in a non-typical manner. Staining with a membrane marker and showing the cell-cell interface in 3D presentation would help.

Response: In order to address this concern from Reviewer 5, we undertook additional coculture experiments wherein we applied the chemical F-actin stain phalloidin (conjugated to Alexa Fluor 488) which stains the cell cortex of adipocytes in addition to several antibody co-stains. Though not a membrane marker, phalloidin was recommended by our coauthors with expertise in adipocyte biology because it stains the adipocyte cortex under the membrane and allows us to distinguish adipocyte and tumor cell interactions. These coculture samples were stained and cleared, then imaged at increased magnification using a 63x immersion objective. (now **Fig 3D** and pasted below for your review). The phalloidin staining helps to better-define the cancer cell membranes and also to visualize the adipocyte borders (particularly at increased contrast — data not shown, **below, bottom**). We now provide these new images in this revision together with our prior images (Fig 3B and Supplementary Fig 3) which show more clearly the association between tumor and adipocytes. We consistently find that TN cells with high Cx31 expression have extensive tumor-adipocyte interaction, while cells in which Cx31 is depleted by CRIPSR engineering or RP cells (T47D) in which expression is low, have much less interaction.

Figure 3d: Staining with Cx31 (magenta), pHSL(S563) (yellow), and phalloidin (blue), of primary mammary tissue from a healthy individual (WD50175) injected with GFP-expressing HCC1143-GJB3^{WT} (top), HCC1143-GJB3^{Low} (middle), or T47D cells (bottom) and co-cultured overnight. White arrowheads indicate Cx31 staining at interface of GFP cancer cells and adipocyte. Scale bar, 10 μ m.

Reviewer #5: The other significant issue not fully addressed is the relative importance of Cx31 exchange of cAMP to adipocyte lipolysis. CBX only caused an apparent 1/3 reduction in dye transfer in Fig. 2C and it is unclear from 3E and G whether the reduction of fluorescence observed with reduced GJB3 levels is 20%, 50% or 80% of the wt level – I understand the levels are significantly reduced but not whether it is compared to the extent of the total dye transfer. The point is that there **seems to be significant residual intercellular communication that exists that should confound the results**. Therefore “Thus, cAMP is transferred from TN breast cancer cells to adipocytes in a Cx31-dependent manner.” is overstated unless you show a zero transfer level upon treatment.

Response: Firstly, regarding cAMP quantification, we were unfortunately unable to measure absolute levels of calcein-AM and fluo-cAMP in adipose tissue from transfer assays presented in Figs. 3E and 3G due to technical limitations of the assay.

Regarding our claims of cAMP transfer being “Cx31-dependent,” we take this to mean that we have not adequately met standards to make this claim. We appreciate Reviewer 5’s rationale here and understand that, in absolute terms this assessment is correct. There are, however, meaningful technical limitations to directly adhering to the presented standard. In theory we would have to utilize an engineered Cx31-null patient-derived TN cell line, but we were notably unable to generate viable total Cx31 knockouts in our patient-derived TN-MYC^{High} cell lines, suggesting that some level of tumor cell expression is necessary to maintain tumorigenic capacity. Moreover, healthy patient breast adipose tissue was utilized in the described coculture assays, and human adipocytes express Cx31; even with Cx31-null TN cells, their hemichannels could still form heterotypic gap junctions with Cx31-containing adipocyte hemichannels.

We do not state that Cx31(GJB3) expression is necessary for cAMP transduction, or that GJB3 gap junctions alone (as opposed to other GJ) can transduce cAMP. In our newly expanded **Discussion** section, we state that “The present data do not, however, exclude transfer of other factors by these GJB3 gap junctions, or transfer of cAMP to adipocytes by other kinds of gap junctions. *In vivo*, we observe that depletion of Cx31 is sufficient to increase retention of cAMP in tumor tissue (**4E** and **below**), and to increase adjacent adipocyte size consistent with diminished transfer of cAMP (**4F** and **below**). In **Figs. 3E and 3G** (and **below**), tumor cell Cx31 depletion is associated with diminished GJ-dependent dye transfer and diminished cAMP transfer to adipocytes *ex vivo*. Prior to Cx31 depletion studies, we also show usage of a moderate concentration of a pan-gap junction inhibitor CBX to first demonstrate that the drug has an effect on GJ-mediated dye transfer (**Fig. 2C** and **below**), and then that those conditions similarly induce cAMP accumulation in TN-MYC^{High} context (**Fig. 3F** and **below**). We believe that these data on partial Cx31 depletion in TNBC, taken together and with consideration of

the technical limitations outlined above, make a meaningful case to claim that in the context of human TN-MYC^{High} tumors and human breast adipocytes, level of cAMP transfer is Cx31-dependent.

If it were technically feasible to fully adhere to the standards of this claim with Cx31-null patient-derived TN tumor cells and adipocytes, we would certainly make every effort; we thank Reviewer 5 for the opportunity to clarify limitations to this statement, and we believe text in the expanded **Discussion** section now appropriately qualifies the claim.

Finally, in the revised manuscript we have removed the specific claim that cAMP transfer is "Cx31 dependent" and instead explain throughout the manuscript that the level of cAMP transfer is correlated to Cx31 expression. We think this most clearly explains the results, especially since complete Cx31 deletion/depletion was not possible.

Fig. 2c Relative frequency of dye transfer from Calcein AM- loaded cells (donor) to unloaded mCherry-labeled cells (recipient) treated with 150uM CBX or vehicle control for 24 hours, as determined by FACS analysis ($p < 0.0001$). Each point represents a biological replicate. **Fig. 3f** cAMP levels in TN high MYC (red), TN low MYC (orange, $p = 0.0487$), and RP (blue, $p = 0.487$) cell lines after treatment with 150 μM CBX for 24 hours, relative to untreated (control) cells. Each point represents a biological replicate averaging three technical replicates. **Fig. 4e** Fold change in cAMP levels in HCC1143 GJB3 Med ($n = 5$) xenografts versus HCC1143 GJB3 WT ($n = 4$) xenografts ($p = 0.0492$). **Fig. 4f** Adipocyte area adjacent to HCC1143 GJB3 Med xenografts (pooled from $n = 5$ tumors, $n = 517$ adipocytes) and HCC1143 GJB3 WT xenografts (pooled from $n = 4$, $n = 771$ adipocytes) and area in control non-tumor (NT) tissue (pooled $n=3$ mice, $n = 2611$ adipocytes) ($p < 0.0001$). The black line indicates mean adipocyte area. Each point represents an individual adipocyte.

Figure 3. e Dye transfer from indicated HCC1143 control and Cx31-depleted lines to primary mammary adipose tissue of indicated ($n = 3$) healthy individuals. **g** cAMP transfer from indicated HCC1143 control and Cx31 partial expression loss lines to primary mammary adipose tissue of indicated ($n = 3$) healthy individuals.

Reviewer #5: Other minor issues

Reviewer #5: “Additionally, in 3 of the 4 models examined we found an increase in HNF4a protein or in phospho-HSL/total HSL ratio (Fig. 1, G and H).” – I see the total protein differences but not significant differences in the pHSL ratio.

Response: In Figure 1G (and pasted below), we probe non-tumor, NAT and tumor tissue from **three distinct** TNBC PDX *in vivo* models (‘HCI100X’); in Figure 1H (and below) we probe the same set of tissues (and mock transplant tissue) in the MTB-TOM MYC-driven breast tumor allograft model. The quoted text references each of these 4 models individually, and for the (3) HCI100X models we only had a single biological tumor lysate sample (such that further western blots or lanes would not have added biological information for statistical evaluation per-model). In the HCI1002 model *and* across MTB-TOM mice (n = 3) we observe **induction of HNF4a in NAT** compared to non-tumor tissue; in the HCI1002 and HCI1010 models we observe **induction of phospho-HSL(S563) normalized to total HSL and then to Actin**. Thus, we state that taken together, in 3 of these 4 examined models we see an increase in HNF4a protein **or** in phospho-HSL/total HSL ratio. We note that final review of our statistical reporting for panel 1G caused us to remove the significance indicator previously shown in this one panel, though the increases by individual tumor model described above and in the text are unchanged. We thank Reviewer 5 for the opportunity here to clarify this statement.

Reviewer #5: In Fig 4 why do the medium Cx31 expressing cells have the most pHSL?

Response: To examine tumor-adipocyte signaling *in vivo* we evaluated several parameters including adipocyte size, tumor cAMP content, and tumor-adjacent pHSL (NAT). Of these parameters, the tumor-adjacent pHSL varied the most amongst the n=3 samples for each condition we tested. Nevertheless, when quantified and normalized to total HSL and then to Actin (Fig 4D), both *GJB3^{WT}* and *GJB3^{Med}* had more pHSL than in the *GJB3^{Low}* samples (**Fig 4D** and pasted **below** for convenience). The difference between *GJB3^{WT}* and *GJB3^{Med}* was not significantly different, although the *GJB3^{Med}* appears highest because of one data point (possibly an outlier) among the three samples. Despite the apparent similarity in pHSL between WT and Med, other parameters indicate more cAMP transfer to adipocytes in *GJB3^{WT}* tumors. In our analysis of intratumor cAMP (**Fig. 4E**; n = 5 mice), we observed a significant increase in *GJB3^{Med}* xenograft tumors compared to *GJB3^{WT}* tumor tissues. We also observed (**Fig. 4F** and pasted below) a concordant significant increase in NAT adipocyte size compared to *GJB3^{WT}* NAT by Adiposoft analysis. Consistent with our proposed mechanism, **Fig 4E** and **4F** suggest that tumor efflux of lipolytic cAMP signaling decreases, and surrounding adipocytes are larger, for Cx31-depleted xenografts. Finally, use of a B3-adrenergic receptor agonist CL316243 in **Fig. 4G** rescues aggressive tumor formation for *GJB3^{Med}* xenografts, indicating that a lipolytic agent may be sufficient to overcome impacts of tumor GJB3 depletion.

Wording issues

Reviewer #5: “We identify connexin 31 (GJB3), which promotes receptor triple negative breast cancer growth and activation of lipolysis *in vivo*.” Weird sentence. It implies the authors were the first to identify Cx31.

Response: We appreciate this comment from Reviewer 5 on wording in this penultimate sentence from the Abstract, which we agree may imply that we were the first to identify Cx31. We have changed this sentence to read, “We find that connexin 31 (GJB3) promotes receptor triple negative breast cancer growth and activation of lipolysis *in vivo*.”

Reviewer #5: “We found a marked positive correlation ($R = 0.5818$, $p = 0.0656$) between the change in lipid content and adipocyte area (Fig. 1D).” You found a trend, not a marked correlation.

Response: We thank Reviewer 2 for their comment, and have amended this sentence to read as follows: “We found a positive *trend* ($R = 0.5818$, $p = 0.0656$) between the change in lipid content and adipocyte area (Fig. 1d).”

Reviewer #5: “In addition, dye uptake in HCC1143 cells was significantly reduced (30.63%, $p < 0.0001$).” Since gap junctions can act as “hemichannels” and uptake dye or other molecules from the media use “transfer” or something else instead.

Response: We thank Reviewer 2 for their input here and the coauthors agree; we have amended this sentence to add greater clarity: “In addition, dye *transfer to* HCC1143 cells was significantly reduced by 30.63% ($p < 0.0001$) following treatment with CBX (Fig. 2c).”

Reviewer #5: In Fig 3H the significant difference comparisons are hard to parse particularly in the HCC1143 low column as is has “ns” under a and apparently other comparisons

Response: We agree with Reviewer 5’s point here that the multiple comparisons were difficult to parse, and thank them for the opportunity to improve panel 3H. We have appended the figure legend to denote use of multiple comparisons 3H, and made a small change to the set of comparisons used in the statistical test to only apply comparisons shown in the panel. The separate, bracketed comparison between HCC1143 GJB3^{WT} or GJB3^{Med} cocultures and the HCC1143 GJB3^{Low} cocultures is meant to indicate that both WT and Med induced greater adipocyte UCP1 expression than GJB3^{Low} cocultures. We note that these specific comparisons to GJB3^{Low} now fall just below significance ($p < 0.10$), and that we now show specific p values for these additional comparisons within the plot. Source data is also available as a Source Data file where we clearly denote the comparisons made and the associated p values.

Reviewer #5: “Adipocytes were then isolated from the tumor cells and assayed for fluo-cAMP. We found that cAMP transfer occurred from control cells to adipocytes from all three patients (Fig. 3G).” Control isn’t very informative here – I presume they mean GJB3WT.

Response: We thank Reviewer 2 for pointing out this potential ambiguity and giving us the opportunity to rectify it. We did intend to mean *GJB3^{WT}*, and have since amended the text to read, “Adipocytes were then isolated from the tumor cells and assayed for fluo-cAMP. We found that cAMP transfer occurred from **control** *GJB3^{WT}* cells to adipocytes from all three patients (Fig. 3g).”

We would like to thank **Reviewer 5** for providing these opportunities to add clarity in the manuscript text. During revisions, we have expanded the Introduction and Discussion sections to add detail and acknowledgement of prior literature and known mechanisms, more thoroughly outlined experimental design and statistical tests, and in response to Reviewer 5’s comments performed new *ex vivo* cocultures with additional cell cortex staining and imaged at higher magnification. We sincerely appreciate the time taken to help us improve upon this study.